# Automated deep-phenotyping of the vertebrate brain

**Amin Allalou[1,2†], Yuelong Wu[1†], Mostafa Ghannad-Rezaie[1,3], Peter M Eimon[1]\*, Mehmet Fatih Yanik[1,3]\***

[1]Massachusetts Institute of Technology, Cambridge, United States; [2]Uppsala University, Uppsala, Sweden; [3]ETH Zürich, Zürich, Switzerland

**Abstract** Here, we describe an automated platform suitable for large-scale deep-phenotyping of zebrafish mutant lines, which uses optical projection tomography to rapidly image brain-specific gene expression patterns in 3D at cellular resolution. Registration algorithms and correlation analysis are then used to compare 3D expression patterns, to automatically detect all statistically significant alterations in mutants, and to map them onto a brain atlas. Automated deep-phenotyping of a mutation in the master transcriptional regulator *fezf2* not only detects all known phenotypes but also uncovers important novel neural deficits that were overlooked in previous studies. In the telencephalon, we show for the first time that *fezf2* mutant zebrafish have significant patterning deficits, particularly in glutamatergic populations. Our findings reveal unexpected parallels between *fezf2* function in zebrafish and mice, where mutations cause deficits in glutamatergic neurons of the telencephalon-derived neocortex.

**\*For correspondence:** peter.
eimon@gmail.com (PME); yanik@
ethz.ch (MFY)

[†]These authors contributed equally to this work

**Competing interests:** The authors declare that no competing interests exist.

## Introduction

Systematic initiatives such as the Zebrafish Mutation Project (ZMP) aim to identify disruptive alleles in all of the more than 26,000 genes in the vertebrate genome and make these available to the scientific community (*Kettleborough et al., 2013*). Additionally, the advent of next-generation genome editing tools is enabling researchers to efficiently target virtually any genomic locus of interest for inactivation or precision alteration. These technologies, combined with prior large-scale mutagenesis screens, have created rapidly growing publicly available collections of zebrafish mutant lines that are in need of detailed characterization. However, rigorously characterizing 1000s of lines quantitatively and in detail remains impractical with current approaches. As a result, phenotyping in large studies is often non-quantitative and restricted to features easily observed by eye. For example, the ZMP characterizes all alleles for ~20 gross structural and behavioral phenotypes (e.g. axis length, movement, etc.) at 5 days post fertilization (dpf) (*Dooley et al., 2013*). Although useful for preliminary screening, these criteria are too general to provide mechanistic insight into alleles that produce obvious defects and completely overlook those with interesting but subtle phenotypes.

Detailed phenotyping, particularly of complex organs like the brain, requires careful characterization of mutant lines using panels of well-established cell- and tissue-specific markers. This can be accomplished either by whole mount in situ hybridization (WISH) using riboprobes or by whole mount immunohistochemistry using antibodies. Most steps of WISH, including probe synthesis and all staining steps, can easily be scaled up and automated using multiwell plates and/or robotic hybridization platforms. However, even in sophisticated large-scale in situ screens, the final crucial steps of documenting and analyzing gene expression patterns are often limited to manual annotation of 2D images acquired from a few standard views (*Antin et al., 2014*; *Quiring et al., 2004*). This is due to the challenges of rapidly imaging complex gene expression patterns in 3D at high-resolution, accurately registering numerous markers across thousands (or more) of samples, and

detecting and quantifying alterations in mutants. Standard 3D imaging modalities [e.g. confocal, two-photon (2P), and selective plane illumination microscopy] require expensive hardware and are restricted to fluorescent readouts. However, gene expression patterns in zebrafish and other model organisms are most commonly studied using WISH protocols employing chromogenic rather than fluorescent readouts. Once 3D images have been acquired, novel computational approaches to phenotyping are essential, since manual analysis suffers from human error (e.g. user fatigue, variation in training and experience) and can only produce semi-quantitative results at best.

To address all of these challenges, we report the first automated scalable phenotyping platform capable of high-resolution isotropic 3D imaging of chromogenic WISH-stained samples along with algorithms for precision registration and quantification of the resultant 3D gene expression patterns. Image acquisition is accomplished using optical projection tomography (OPT), a 3D imaging technique operating on the same principle as X-ray computed tomography but utilizing visible radiation instead of X-rays (*Reynaud, 2013*; *Sharpe et al., 2002*). OPT is capable of imaging both absorption of transmitted light and emitted fluorescence (*Wang and Wang, 2007*), making it ideal for deep-phenotyping of chromogenic WISH. Compared to fluorescence imaging techniques, acquisition times for OPT are extremely short when high-contrast stains are used, a crucial requirement for large-scale analysis. We have previously described a high-throughput OPT system to quantify craniofacial skeletons in zebrafish embryos using the Alcian blue histological stain (*Pardo-Martin et al., 2010*, *2013*). In the present paper, we adapt our OPT platform to WISH-stained samples and develop a suite of algorithms that automatically registers expression patterns to a common reference frame with a high level of precision. This enables us to detect statistically significant alterations between wild-type and mutant embryos on a voxel-by-voxel basis using automated correlation analysis and map these defects to an anatomical atlas. Applying this analysis to a diverse library of brain-specific riboprobes allows us to detect and quantify even subtle deficits in mutant embryos in an unbiased manner. Additionally, because all 3D patterns are registered to a common frame, our platform allows any desired co-expression pattern to be easily visualized without the need for double- or triple-labeling experiments. For a 20-probe WISH library, this means that all unique 3-probe combinations can be analyzed by performing just 20 single-label in situ hybridizations rather than 1140 triple-label hybridizations. Our new high-throughput OPT system can be assembled inexpensively using off-the-shelf components and enables 3D imaging of non-embedded WISH-stained samples at cellular resolution in less time than conventional 2D imaging, resulting in an orders-of-magnitude increase in data from standard in situ experiments.

To demonstrate the ability of automated 3D phenotyping to detect important but easily overlooked defects, we analyzed zebrafish embryos containing a previously characterized mutation in *fez family zinc finger 2* (*fezf2*; OMIM *607414). *Fezf2* is an evolutionarily conserved transcription factor that plays critical roles in neurogenesis and cell fate specification in organisms ranging from *Drosophila* to mammals (*Eckler and Chen, 2014*). In mice, *Fezf2* is expressed in forebrain neural progenitors and postmitotic glutamatergic projection neurons in layers 5 and 6 of the neocortex (*Inoue et al., 2004*). Knockout mice show a striking loss of corticospinal motor neurons and other closely related subcerebral projection neurons in cortical layer 5, which are replaced by callosal projection neurons (*Molyneaux et al., 2005*; *Chen et al., 2005a*, *2005b*, *2008*). Although developmental expression of *Fezf2* is similar in fish and mammals (*Berberoglu et al., 2009*; *Blechman et al., 2007*; *Levkowitz et al., 2003*; *Yang et al., 2001*; *Chen et al., 2005a*, *2005b*; *Hirata et al., 2004*; *Molyneaux et al., 2005*), previously reported loss-of-function phenotypes differ in important aspects. The function of *fezf2* in zebrafish has been extensively studied using the *fezf2^m808* mutant (also known as *too few*), which is thought to be a hypomorphic loss-of-function allele (*Jeong et al., 2006*). In contrast to knockout mice, where *fezf2* defects are seen in the glutamatergic projection neurons of the telencephalon-derived neocortex, early developmental deficits in zebrafish embryos have only been reported in diencephalon, particularly in dopaminergic (DA), serotonergic (5-HT), oxytocinergic-like, and GABAergic populations (*Blechman et al., 2007*; *Rink and Guo, 2004*; *Yang et al., 2012*; *Guo et al., 1999*; *Levkowitz et al., 2003*).

These differences raise the possibility that *fezf2* function has diverged substantially over time, particularly in regions of the telencephalon giving rise to the mammalian neocortex. Alternatively, important but subtle aspects of the *fezf2* phenotype may have been overlooked during zebrafish development. Support for the latter possibility comes from the finding that adult *fezf2* mutant zebrafish have significantly smaller telencephalons, in spite of the fact that early developmental defects

are reported in the diencephalon (*Berberoglu et al., 2014*). Although the telencephalon appears grossly normal at two weeks post fertilization, we speculated that deep-phenotyping might uncover earlier abnormalities. We therefore analyzed *fezf2* mutants at two and three days post fertilization (dpf) using a diverse in situ riboprobe library to detect progenitor populations, differentiated neuron subtypes, and brain regions. Our custom image analysis algorithms allowed us to automatically detect and quantify areas of significantly altered gene expression throughout the entire brain. In addition to previously reported defects in the ventral diencephalon, we found a second phenotypic hotspot in the telencephalon. Detailed analysis shows that *fezf2* mutants exhibit a variety of novel forebrain abnormalities as early as 2 dpf. Notably, we report for the first time that mutants display a dramatic loss of glutamatergic neurons in the pallium of the telencephalon, although telencephalic DA and GABAergic populations appear to be relatively normal. This phenotype is accompanied by a reduction in telencephalon volume. Our findings show that the zebrafish *fezf2* phenotype corresponds much more closely to mammalian deficits than previously assumed. Our data suggest a model in which *fezf2* controls two distinct developmental programs in the zebrafish brain: (1) a glutamatergic program in the telencephalon with similarities to mammalian *Fezf2* function, and (2) a broader neurogenic program in the diencephalon.

## Results

### High-resolution 3D imaging of gene expression patterns using high-throughput OPT

We have previously demonstrated a high-throughput OPT platform for rapid 3D imaging of non-embedded zebrafish embryos at micrometer resolution (*Pardo-Martin et al., 2013*, *2010*). Because formaldehyde fixation renders normally transparent zebrafish embryos optically dense, high-resolution OPT requires tissue clearing techniques to reduce light scattering and maximize transparency. In our previous publication, clearing was achieved using a trypsin solution on embryos stained with the histological dye Alcian blue. Since Alcian blue is limited to detecting cartilage, we sought to make OPT applicable to all anatomical structures or genes of interest by adapting our platform to embryos stained using chromogenic WISH. This required an alternative tissue clearing technique, as trypsin is ineffective on WISH-stained samples. Immersion in a mixture of benzyl alcohol and benzyl benzoate (BABB) is a simple and rapid solvent-based clearing technique. BABB's hydrophobicity and high refractive index (RI; 1.559) render samples optically transparent, making deeper structures accessible for high-resolution 3D imaging (*Becker et al., 2012*). The switch to BABB necessitated several changes to our OPT platform. In order to match the RI of BABB, we load samples into an aluminosilicate glass capillary (RI = 1.538) instead of a borosilicate capillary (RI = 1.474), which is immersed in a BABB-filled glass cuvette to achieve RI matching between the inside and outside (*Figure 1A*; details in Materials and methods). We also developed a process for fabricating a tapered transparent insert from RI-matched optical adhesive within the capillary to accommodate embryos from multiple developmental stages and ensure stability during high-speed rotational imaging (*Figure 1B*, *Figure 1—figure supplement 1A–C*; details in Materials and methods). The capillary is mounted vertically within the imaging chamber, allowing rapid loading and unloading through a single fluidic port at the top and further enhancing stability during rotation.

Once a WISH-stained embryo has been introduced into the capillary, it is rapidly rotated through 360°, allowing images to be continuously acquired at a high frame-rate from multiple angles. For each specimen, we acquire 360 color images with a pixel resolution of ~1.5 µm and a rotation time of ~12 s (*Figure 1—figure supplement 1D–I*). Following image acquisition, individual raw images are aligned using methodology and algorithms we developed previously (see Materials and methods) (*Pardo-Martin et al., 2013*). A GPU implementation of the Filtered back projection from the ASTRA Tomography Toolbox (*van Aarle et al., 2015*, *Palenstijn et al., 2011*) is used for the tomographic reconstruction. Resulting 3D images can be visualized from any angle or plane of section, segmented, and analyzed using the deep-phenotyping approaches discussed below (*Figure 1C,D*; *Video 1*). All alignment and reconstruction steps are automated and can be run in parallel to increase throughput.

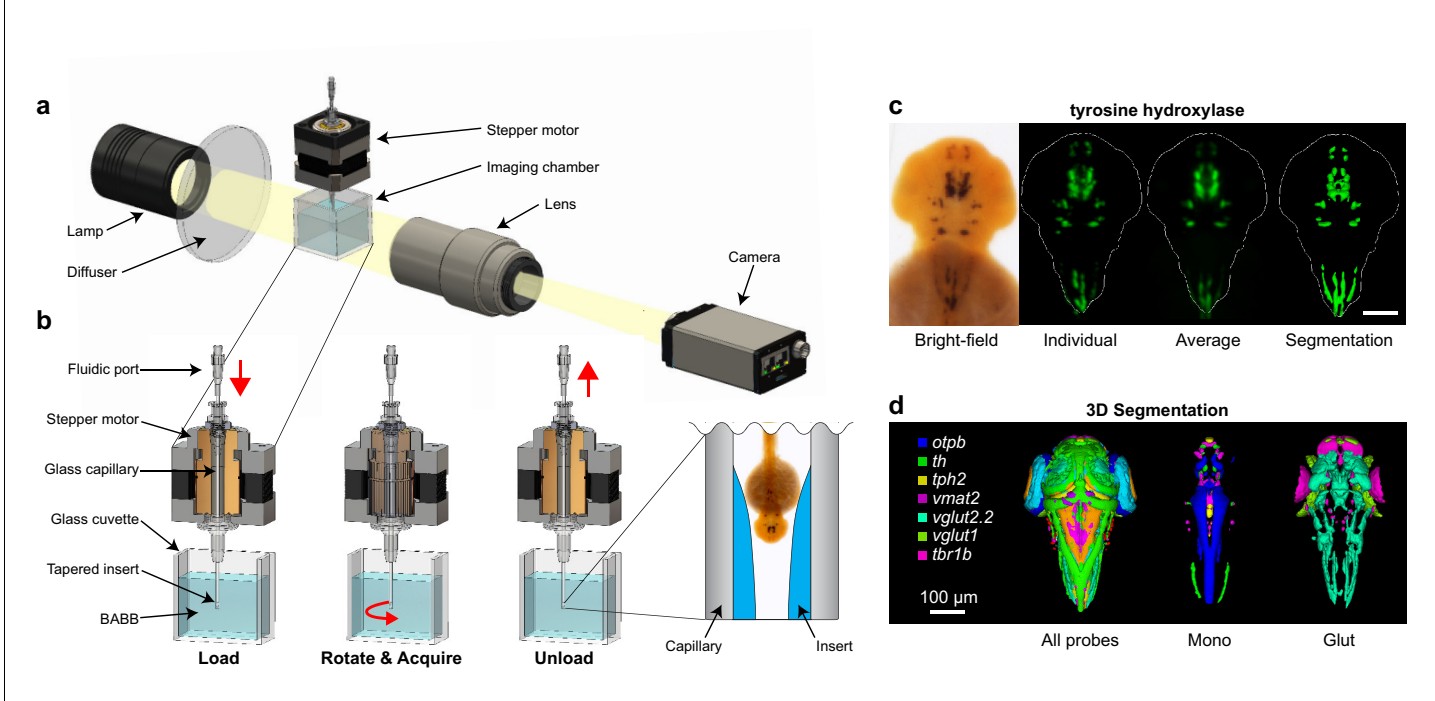

**Figure 1.** Automated OPT platform for automated 3D in situ phenotyping. (**A**) From left to right, the optical projection tomography platform consists of the following components: (1) a post-mountable broadband emission quartz tungsten-halogen light source, (2) a ground glass diffuser, (3) an imaging chamber and glass capillary capable of rotating the specimen 360°, (4) a telephoto zoom lens, and (5) a programmable progressive scan CCD camera (one megapixel, 120 fps). (**B**) The imaging chamber consists of an upper hollow shaft stepper motor run by 5-phase microstepping drivers and a lower transparent glass cuvette. The stepper motor holds and rotates an aluminosilicate glass capillary containing a non-embedded, paraformaldehyde-fixed zebrafish embryo stained using standard chromogenic whole-mount in situ hybridization techniques. Both the capillary and the surrounding glass cuvette are filled with benzyl alcohol-benzyl benzoate (BABB) to achieve refractive index matching and render the fixed sample optically transparent. The bottom end of the capillary contains a tapered insert fabricated from index-matching optical adhesive to hold the sample in place during rotational imaging and enable rapid loading and unloading through the upper fluidic port. The location and movement of the embryo during each step is indicated by the red arrow. (**C**) Tomographic reconstruction of a 2 dpf embryo stained with *tyrosine hydroxylase* showing from left to right: a typical single 2D bright-field image, a 3D reconstruction of an individual embryo presented as a maximum intensity projection, an averaged 3D expression pattern based on 10 embryos, and a 3D segmentation of the averaged pattern. (**D**) Co-registered 3D segmentations of all probes in our library (left), monoaminergic markers (mono, center), and glutamatergic markers (glut, right). (**C,D**) Anterior is to the top. Scale bar: 100 μm.

The following figure supplements are available for figure 1:

**Figure supplement 1.** Capillary fabrication and OPT image acquisition.

**Figure supplement 2.** Registering 3D zebrafish images.

**Figure supplement 3.** Registration workflow for in situ pattern alignment.

**Figure supplement 4.** alternate registration workflow for in situ pattern alignment.

## Registration and alignment of 3D gene expression datasets

3D images must be registered to a common reference frame for comparison of gene expression between multiple probes or experimental groups. Automated algorithms for global registration of 3D fluorescent images from zebrafish have been described (*Randlett et al., 2015*; *Ronneberger et al., 2012*), however these are unsuitable for tomographic reconstructions from chromogenic WISH as they require imaging two stains in separate fluorescence channels: (1) a detailed marker of internal morphology, which is used to stain all fish and guides deformable registration to the common reference frame and (2) the specific expression pattern of interest. We therefore devised a novel approach using MATLAB's Image Processing Toolbox and the open source

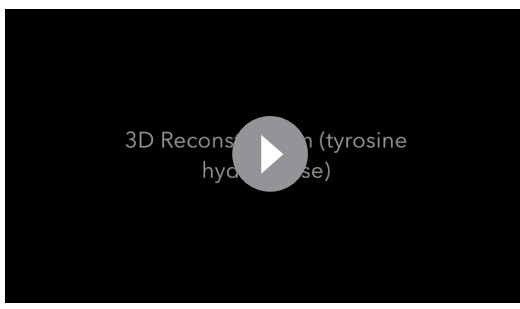

**Video 1.** 3D reconstruction of a wild-type 2 dpf embryo stained with tyrosine hydroxylase.

registration toolbox elastix (*Ibanez et al., 2005*; *Shamonin et al., 2013*; *Klein et al., 2010*). Our registration strategy is described in detail in Materials and methods and utilizes both the dark blue chromogenic in situ stain (which can be clearly visualized in the red or green channels of an RGB image and highlights internal morphology) and a faint red SYTOX Green counterstain (which can be visualized in the blue or green channels and highlights surface features that are common to all embryos). We begin by creating an average 3D unstained reference fish (URF) for each developmental stage to serve as a common reference frame for global registration. This is done using iterative shape averaging (ISA) (*Rohlfing et al., 2001*) in the green channel on multiple embryos that have been processed using the standard WISH protocol with the riboprobe omitted. All WISH-stained embryos are then aligned to the appropriate age-specific URF as detailed in Materials and methods. Briefly: for each probe a single stained embryo is selected as a Probe Reference Fish (PRF). The PRF (blue channel) is registered to the URF (green channel) using the SYTOX stain and all other embryos stained with the same probe are registered to the PRF using both the SYTOX stain and the 3D gene expression pattern (green channel). Registrations are done with a non-deformable registration followed by a deformable registration (*Figure 1—figure supplements 2* and *3*; details in Materials and methods). Following registration, users can view either 3D reconstructions of single embryos or averaged 3D expression patterns from multiple embryos stained with the same probe. The latter option compensates for biological variation and is particularly attractive for probes with diffuse or punctate expression patterns [e.g. *tyrosine hydroxylase* (*th*), *Figure 1C*]. We have therefore used it for all subsequent experiments unless noted.

We quantified the accuracy of our registration algorithms using *tryptophan hydroxylase 2* (*tph2*). This probe labels discrete expression domains in the epiphysis and the raphe serotonergic neurons and shows relatively little embryo-to-embryo variation in size or shape, making it an ideal landmark for assessing alignment accuracy (*Figure 2A*). We stained 26 wild-type embryos from the same clutch, randomly divided them into three groups, and independently registered each group using our algorithms. Both *tph2* expression domains were then segmented in all embryos (see Materials and methods). In order to quantify misalignment, we measure how much the location of each domain deviates between pairs of aligned embryos using a border distance measure (*Tsai et al., 2003*). This is done by calculating the minimum distance from each border voxel of the domain segmentation in the first embryo to the border of the same domain segmentation in the second embryo. The final border voxel distance for an embryo pair is the average of all minimum distances. We then compare border voxel distances from all pairs within the same registration group (intragroup) and from all pairs in different registration groups (inter-group) (*Figure 2B*). Our algorithms produce alignments that are accurate to sub-cellular scales for both the raphe expression domain (intra-group registration accuracy = 3.5 μm; inter-group accuracy = 4.0 μm) and the epiphysis expression domain (intra-group accuracy = 2.3 μm; inter-group accuracy = 2.5 μm). To further validate alignment accuracy, we examined genes with well-characterized co-expression in zebrafish. *Vesicular monoamine transporter 2* (*vmat2*), *th*, and *tph2* are expressed in monoaminergic, catecholaminergic, and 5-HT neurons respectively. In zebrafish, *th* is expressed in DA neurons of the telencephalon and diencephalon and noradrenergic (NA) neurons of the locus coeruleus (*Rink and Wullimann, 2002*). *Vmat2* is expressed in DA neurons of the telencephalon and diencephalon, NA neurons of the locus coeruleus, and 5-HT neurons of the raphe nuclei (*Wen et al., 2008*). *Tph2* is expressed in 5-HT neurons of the epiphysis and the raphe nuclei (*Teraoka et al., 2004*). When we examine 2D slices from 3D reconstructions of *th*, *vmat2*, and *tph2* that have been co-registered using our algorithms, all sites of co-expression (*th* and *vmat2* in DA clusters; *th* and *tph2* in the locus coeruleus and raphe nuclei) are accurately recapitulated (*Figure 2C*). Taken together, these results demonstrate our OPT platform and algorithms are highly robust and allow 3D WISH patterns to be reliably compared between separately stained embryos.

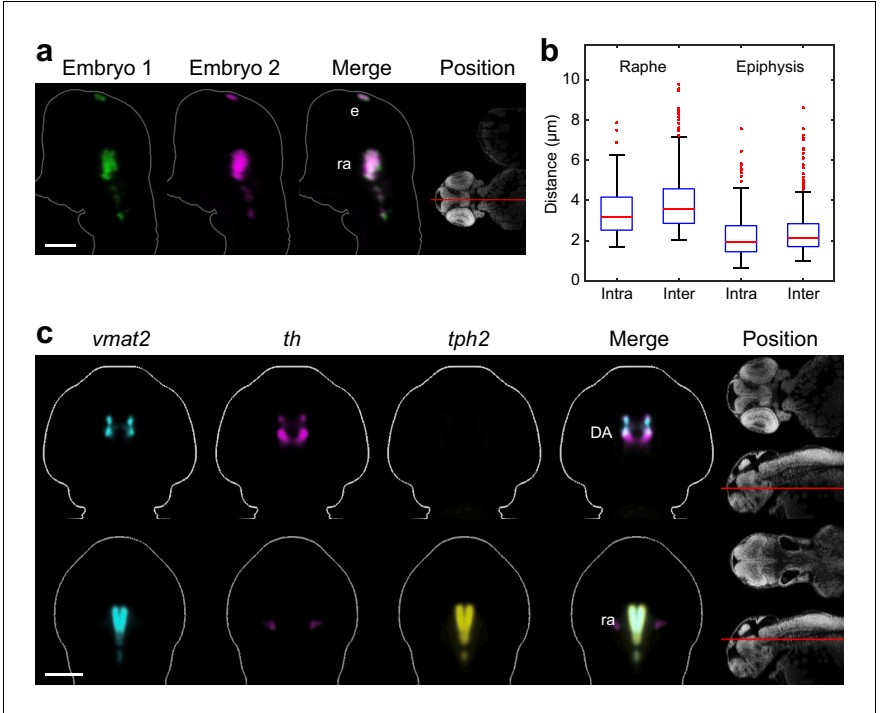

**Figure 2.** Automated registration and alignment of 3D WISH images. (**A**) Alignment accuracy of 3D registration algorithms was verified using 26 wild-type embryos stained with *tryptophan hydroxylase 2* (*tph2*). Stained embryos are randomly divided into three groups and all groups are independently registered to the common reference. The overlay analysis shows two representative embryos from separate registration groups. The position and orientation of each 2D slice within the embryo is indicated on the Nissl-stained two photon reference image to the right. Anterior is to the top. (**B**) Registration accuracy was quantified by manually segmenting *tph2* expression domains in all three independent groups and calculating the average distance between every border voxel in the segmentation. For the hindbrain expression domain in the raphe nuclei (ra), the intra-group registration accuracy is 3.5 μm and the inter-group accuracy is 4.0 μm. For the epiphysis (ep) expression domain, the intra-group accuracy is 2.3 μm and the inter-group accuracy is 2.5 μm. Box-and-whisker plots show results of intra- and inter-group border distance measurements. Tops and bottoms of each box represent the 25th and 75th percentiles of the samples, respectively. Whiskers are drawn from the ends of the interquartile ranges to the furthest observations that fall within ±1.5 times the interquartile range away from the top or bottom of the box. The line in the middle of each box is the sample median. Observations beyond the whisker length are marked as outliers (+ sign). (**C**) Virtual co-registration of *vesicular monoamine transporter 2* (*vmat2*), *tyrosine hydroxylase* (*th*), and *tryptophan hydroxylase 2* (*tph2*) in wild-type embryos at 2 dpf showing patterns of colocalization in the raphe nuclei and dopaminergic (DA) clusters. All 3D reconstructions are generated by averaging eight or more embryos per experimental group. For transverse planes (upper) dorsal is to the top; for frontal planes (lower) anterior is to the top. The position and orientation of each 2D slice within the embryo is indicated on the Nissl-stained two photon reference image to the right of the panel. Scale bar: 100 μm.

To enable quantification and analysis of gene expression based on anatomical context, we created 3D reference atlases for zebrafish brains at 2 and 3 dpf (Source code 1). Atlases were developed from 3D images generated by 2P excitation microscopy using embryos stained with green fluorescent Nissl stain. This histological stain is widely used to study neuromorphology (*Nissl, 1894*; *Kádár et al., 2009*) and provides a convenient way to visualize general brain anatomy. Embryos were counterstained with red fluorescent nucleic acid stain, giving uniform background labeling and enabling registration to age-specific URFs. To achieve the best match between anatomical atlases and OPT reconstructions, embryos were processed using the WISH protocol without a riboprobe and immersed in BABB during 2P microscopy. Anatomical segmentation was done manually following established zebrafish brain nomenclature and with the aid of the Atlas of Early Zebrafish Brain Development (*Mueller and Wullimann, 2015*) and the Virtual Brain explorer 3D (ViBE-Z) atlas

(*Ronneberger et al., 2012*). Segmentations were performed using TurtleSeg 1.3, an interactive segmentation tool developed for 3D medical images (*Top et al., 2011*, *Top et al., 2010*) (*Figure 3—figure supplement 1*). Following registration, 3D atlases can be used to automatically map expression intensity and volume to brain regions (*Ronneberger et al., 2012*) (*Figure 3—figure supplement 2*) or for deep phenotypic analysis, as discussed below.

## Automated deep-phenotyping of entire brains in zebrafish mutants

The ability to rapidly image and align thousands of WISH-stained embryos in 3D enables automated phenotyping using unbiased quantitative algorithms. To demonstrate the power of this approach, we performed deep-phenotyping on a loss-of-function mutation in the *fezf2* transcription factor, which is essential for neurogenesis in many organisms (*Guo et al., 1999*). Phenotyping was done using a diverse library of brain-specific riboprobes to visualize a range of progenitor and postmitotic populations throughout the brain (*Supplementary file 1*). For each probe, we imaged 8+ wild-type and mutant embryos at 2 and 3 dpf. In order to automatically detect regional differences in expression, we developed an Automated Correlation Analysis framework based on an approach used for group comparisons in brain network studies (*Simpson et al., 2013*). Following registration, we calculate a Pearson correlation coefficient between each wild-type and mutant embryo for every probe within all brain regions based on the intensities of corresponding voxels (*Goshtasby, 2012*). For each brain region, a correlation matrix is generated containing intra-group correlations for wild-types and mutants and inter-group correlations between wild-types and mutants (*Figure 3A*, right). If there is a phenotypic difference, the intra-group correlation is expected to be significantly higher than the inter-group correlation. We used false discovery rate (FDR) estimation, a technique for assessing false positives when conducting multiple comparisons, to set an appropriate *p*-value threshold for automated phenotyping (*Noble, 2009*). To estimate FDR, we compare our experimental data set (correlation between wild-type and *fezf2* mutants) with a null data set consisting of negative control embryos (i.e. embryos that are either genetically wild-type or heterozygous for the *fezf2* mutation) that have been randomly divided into two separate groups following in situ staining (*Figure 3—figure supplement 3A*). Since negative control embryos lack the *fezf2* phenotype, any differences that are detected between the two null groups are presumed to be false positives. Based on this analysis, we have set our *p*-value threshold at $p < 10^{-5}$, which gives a FDR of 0.043. In addition to significance testing, we calculate the effect size (Cohen's *d*) to measure the magnitude of the difference in correlation (*Sullivan and Feinn, 2012*). In the null data set, the mean effect size is less than 1.0 for all significance thresholds evaluated (*Figure 3—figure supplement 3A*). This suggests that small effect sizes are indicative of minor changes most likely arising from differences in background staining or biological variation, while larger values indicate true alterations in gene expression. Therefore, we included an effect size requirement ($d > 1.0$) for Automated Correlation Analysis, further reducing the FDR at our selected *p*-value threshold from 0.043 to 0.020 (*Figure 3—figure supplement 3B*). These criteria ($p < 10^{-5}$, $d > 1.0$) are used to create an Automated Correlation Analysis table depicting changes in all probes across all brain regions at 2 and 3 dpf (*Figure 3A*, left).

Automated Correlation Analysis serves as the basis for additional deep phenotyping approaches that highlight various features of interest in 3D WISH datasets. To better detect phenotypic hotspots in which multiple probes are altered, we calculate the voxel intensity difference between wild-types and mutants for each probe within each brain region that shows a significant change based on Automated Correlation Analysis. Summing the absolute value of the differences for all probes results in an aggregated difference image that highlights the voxels that are most altered in *fezf2* mutants. Aggregated difference images clearly show two phenotypic hotspots in *fezf2* mutants: one in the ventral diencephalon and one in the telencephalon (*Figure 3B*, *Video 2*). To visualize whether expression is increased, decreased, or altered in more complex ways, we perform a Mann-Whitney *U* test to compare corresponding voxels in wild-types and mutants for each probe within each brain region that shows alterations in Automated Correlation Analysis. This allows us to determine if there is a significant increase or decrease in expression on a voxel-by-voxel basis. As with Automated Correlation Analysis, we use FDR estimation and our null data set to select an appropriate *p*-value threshold for voxelwise analysis and to verify the accuracy of our approach (*Figure 3—figure supplement 4*). In order to visualize significant alterations, we utilize maximum intensity projections (MIPs) color coded to highlight regions in which expression is significantly reduced (cyan) or increased (magenta) in mutant embryos (*Figure 3C*, *Supplementary file 2*).

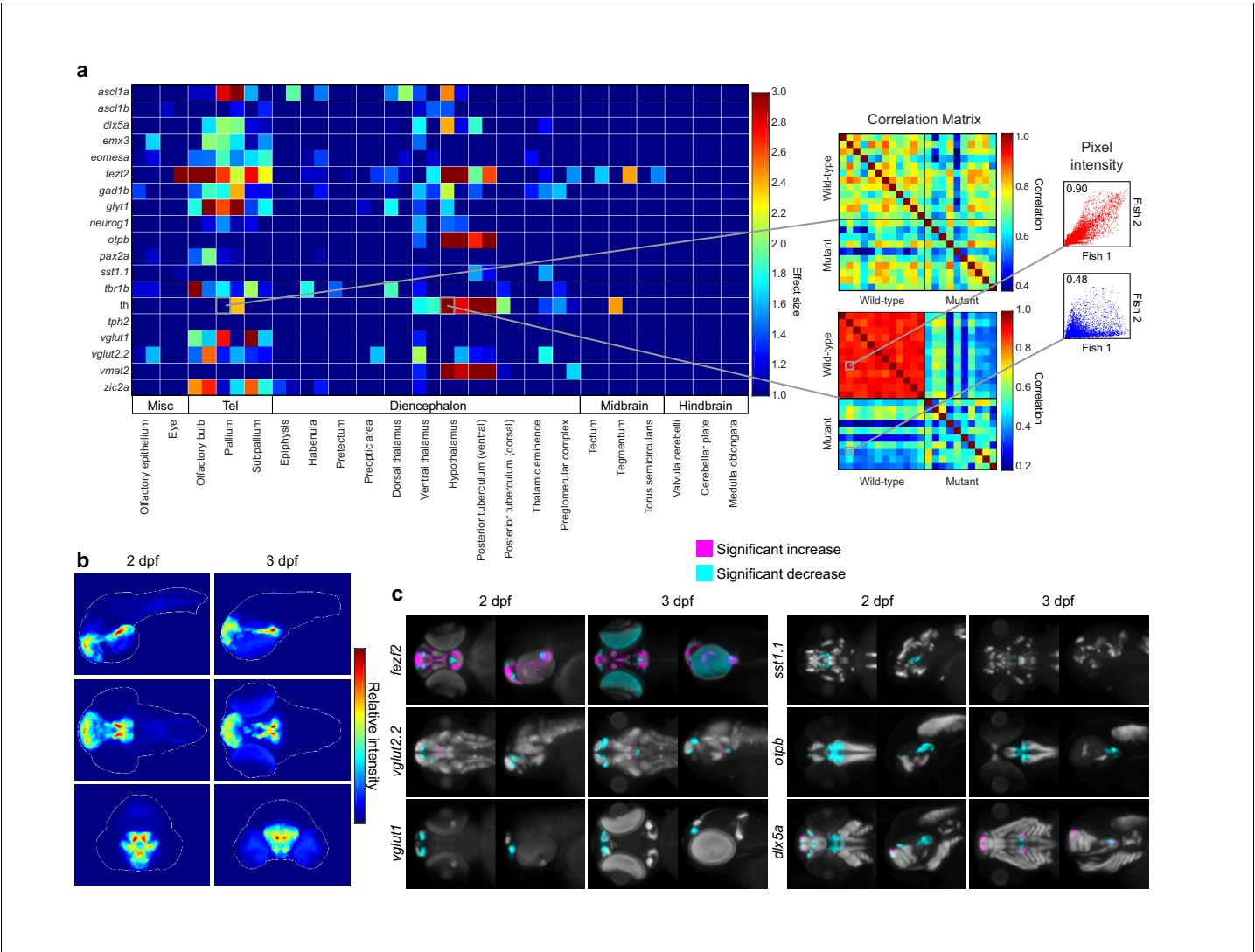

**Figure 3.** Automated detection and statistical quantification of *fezf2* mutant deficits. (A) Right: alterations in the expression of individual probes can be detected using a correlation-significance analysis approach. For each brain region, all wild-type embryos are compared with each other and with all mutant embryos, creating a correlation matrix. The Pearson correlation coefficient is calculated for each embryo pair based on the intensities of corresponding voxels. For probes with altered expression patterns in *fezf2* mutants, correlation within groups (i.e. wild-type vs. wild-type and mutant vs. mutant) is significantly greater than correlation between groups. Scatterplots to the right of the correlation matrix illustrate embryo pairs with correlated (red; $r = 0.9$) and uncorrelated (blue; $r = 0.5$) expression. Left: Automated Correlation Analysis plot showing results for all in situ probes in all brain regions at 2 and 3 dpf. For each region, 2 dpf results are shown in the left square and 3 dpf in the right. Dark blue squares indicate no significant difference between wild-type and mutant embryos ($p<10^{-5}$). For regions with a significant difference, the color bar represents the effect size ($d > 1$) of the difference. (B) Regions in which the expression patterns of multiple independent probes have been altered can be visualized using aggregate difference imaging. Pixel intensity difference is calculated between wild-types and mutants for each probe within each affected brain region. Summing the absolute value of the significant differences for all probes highlights areas which are most altered in *fezf2* mutants. Aggregate difference images are shown at 2 and 3 dpf in lateral (upper panels), dorsal (middle panels), and anterior (lower panels) views. Telencephalon, t; ventral diencephalon, vd. (C) Significance analysis overlaid on maximum Intensity Projections (MIPs). MIPs have been color coded to highlight all regions in which the expression of a given probe is significantly reduced (cyan) or increased (magenta) in mutant embryos ($p<0.5\times10^{-3}$). Significant intensity differences between mutants and wild-types were determined by performing a Mann-Whitney *U*-test to compare corresponding voxels for each probe within each brain region showing alterations in Automated Correlation Analysis. All MIPs show dorsal (left) and lateral (right) views. (B,C) Anterior is to the left, dorsal is to the top.

The following figure supplements are available for figure 3:

**Figure supplement 1.** 3D anatomical brain atlases.

*Figure 3 continued on next page*

**Figure supplement 2.** Expression data mapped to brain regions.
DOI: 10.7554/eLife.23379.011
**Figure supplement 3.** Negative controls for automated 3D phenotyping.
DOI: 10.7554/eLife.23379.012
**Figure supplement 4.** Negative controls for maximum intensity projections.
DOI: 10.7554/eLife.23379.013

## Automated 3D phenotyping detects known and novel diencephalic deficits in *fezf2* mutants

One of the phenotypic hotspots detected by Automated Correlation Analysis is in the ventral diencephalon, an area encompassing the preoptic area, ventral thalamus, hypothalamus, and ventral posterior tuberculum (*Figure 3A,B*). This agrees with previous reports that *fezf2* is essential for multiple neuronal subtypes in the ventral diencephalon, including DA (*Rink and Guo, 2004*; *Levkowitz et al., 2003*; *Guo et al., 1999*), 5-HT (*Levkowitz et al., 2003*; *Rink and Guo, 2004*), and GABAergic neurons (*Yang et al., 2012*). Our analysis clearly picks up several established deficits. We see significantly reduced correlation in one or more ventral diencephalic regions for the GABAergic markers *distal-less homeobox 5a* (*dlx5a*) and *glutamate decarboxylase 1b* (*gad1b*) and the DA markers *th*, *vmat2*, and *orthopedia homeobox b* (*otpb*) (*Figure 3A*). MIPs (*Figure 3C*, *Supplementary file 2*) and virtual 2D slices (*Figure 4A,B,E*, *Figure 4—figure supplement 1A–E*) show that these probes are significantly reduced in the ventral diencephalon of *fezf2* mutants. Loss of DA markers is seen at both 2 and 3 dpf, while *dlx5a* and *gad1b* appear to partially recover by 3 dpf (*Supplementary file 2*, *Figure 4—figure supplement 1A,B*), suggesting for the first time that the GABAergic phenotype in *fezf2* mutants may be transient. Although our analysis did not detect defects in 5-HT neurons, this is likely because our only 5-HT-specific probe (*tph2*) is not expressed in the ventral diencephalon during the stages analyzed.

Most *fezf2* diencephalic deficits are restricted to a subset of the expression domains for a given probe. For example, although *otpb* (a homeodomain protein essential for specification of subsets of neuroendocrine cells and DA neurons) is lost in the ventral posterior tuberculum and posterior hypothalamus, it remains largely unchanged in the anterior hypothalamus and other regions of the diencephalon (*Figure 4—figure supplement 1D,G,H*). Similarly, *th* expression is significantly reduced in the DA clusters of the ventral posterior tuberculum and hypothalamus, with posterior clusters appearing to be more strongly affected than anterior clusters (*Figure 4—figure supplement 1C*). These subdomain-specific defects in *otpb* and *th* have been characterized in detail in *fezf2^{m808}* mutants (*Rink and Guo, 2004*; *Blechman et al., 2007*), providing validation for our approach to automated phenotyping.

In addition to detecting previously reported diencephalic phenotypes, we uncovered several overlooked yet highly significant deficits in *fezf2* mutants. The early developmental expression pattern of *fezf2* has been reported to be normal in *fezf2^{m808}* mutants (*Levkowitz et al., 2003*), but our analysis shows significant alterations in the ventral diencephalon and other regions (*Figure 3A*). Visualization of *fezf2* using MIPs reveals that virtually all expression domains are altered (*Figure 3C*). No other probe in our library shows such widespread disruption (*Supplementary file 2*), suggesting *fezf2* plays a critical role in regulating its own expression either directly or via feedback from downstream transcriptional targets. To better understand this phenotype, we examined virtual 2D slices in

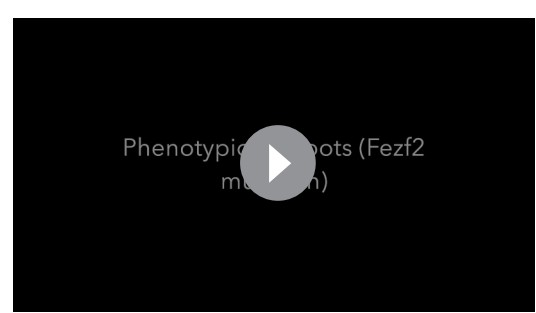

**Video 2.** 3D visualization of aggregate difference imaging at 2 days post fertilization. Voxel intensity difference is calculated between wild-types and *fezf2* mutants for each probe within each affected brain region. Summing the absolute value of the significant differences for all probes highlights phenotypic hotspots where multiple gene expression patterns are disrupted in mutant embryos.

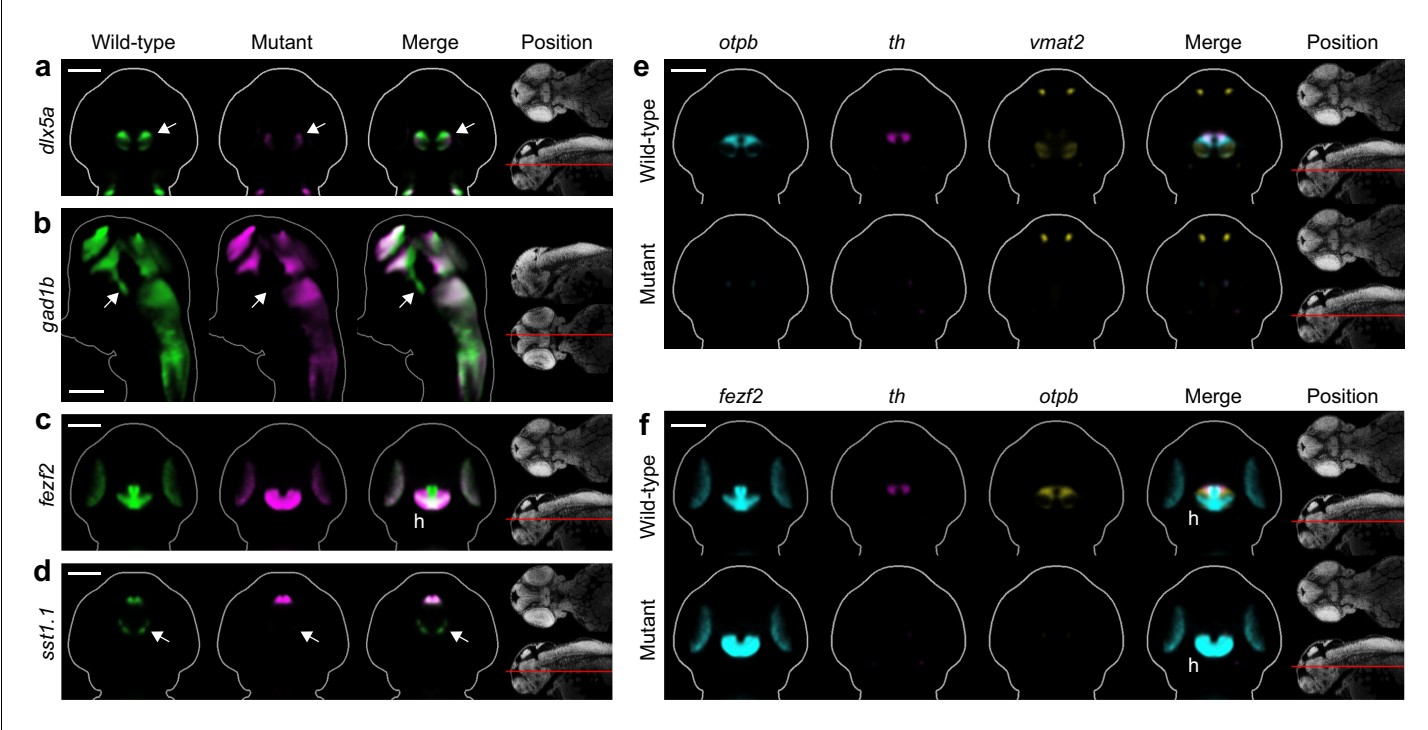

**Figure 4.** Automated phenotyping uncovers known and novel diencephalic deficits in *fezf2* mutants. (A–D) Overlay analysis of in situ expression patterns in wild-type and *fezf2* mutant embryos at 2 dpf. Wild-type expression patterns are shown in green and *fezf2* mutants are shown in magenta. Expression of the GABAergic markers *dlx5a* (A) and *gad1b* (B) is substantially reduced (arrow) in the ventral diencephalon. (C) The *fezf2* expression domain is dramatically altered in the posterior hypothalamus (h). (D) A bilateral cluster of *sst1.1*-expressing cells (arrow) is lost in the ventral diencephalon. Other clusters are unaffected. (E–F) Multi-probe co-expression analysis of ventral diencephalic deficits in *fezf2* mutants. (E) Virtual co-registration showing loss of *otpb*, *th*, and *vmat2* expression domains in the ventral diencephalon. (F) Virtual co-registration showing spatial relationships between affected *fezf2*, *th*, and *otpb* expression domains in the posterior hypothalamus. (A–F) All 3D reconstructions are generated by averaging eight or more embryos per experimental group. The position and orientation of each 2D slice within the embryo is indicated on the Nissl-stained two photon reference image to the right of the panel. For sagittal and frontal sections anterior is to the top; for transverse sections dorsal is to the top. Scale bar: 100 μm.

The following figure supplement is available for figure 4:

**Figure supplement 1.** Additional diencephalon phenotypes in *fezf2* mutant embryos.

wild-type and mutant embryos. In the posterior hypothalamus of wild-type embryos, *fezf2* is expressed in a T-shaped pattern running along the midline and spreading into the lateral hypothalamus (*Figure 4C*, *Figure 4—figure supplement 1F*). When *otpb* and *th* are virtually co-registered with *fezf2*, they are located in an anterior/lateral position relative to *fezf2* (*Figure 4F*). This distinctive arrangement of *fezf2* and *otpb* in the posterior hypothalamus has been described previously using two-color fluorescent in situ hybridization (*Wolf and Ryu, 2013*). In *fezf2* mutants, *fezf2* expression is lost along the midline while simultaneously expanding laterally and anteriorly into regions normally expressing *otpb* and *th* (*Figure 4F*). Previous work shows that *fezf2* is required for expression of *otpb* and the closely related *otpa*. *Otp* genes in turn act as negative regulators of *fezf2* (*Yang et al., 2012*; *Wolf and Ryu, 2013*). Taken together with our observations, this suggests *fezf2* could be transiently expressed in anterior/lateral regions of the hypothalamus that eventually become *otp* positive and *fezf2* negative. In wild-type embryos, negative feedback from *otp* may be required to turn off *fezf2* transcription in these regions and create distinct non-overlapping expression domains. In mutants, the non-functional Fezf2 protein would be unable to initiate *otp* expression, allowing *fezf2* expression to persist. Since *otp* genes are essential for development of DA neurons in the diencephalon (*Fernandes et al., 2013*; *Ryu et al., 2007*), it is not surprising that *th* expression is lost in this region of *fezf2* mutants.

Automated phenotyping also shows that somatostatin-expressing neurons (*sst1.1*) are lost in the ventral diencephalon. Although distinct clusters of *sst1.1* neurons are present in many regions, deficits in *fezf2* mutants are limited to a small bilateral cluster located between the hypothalamus and the ventral thalamus/ventral posterior tuberculum (*Figures 3C* and *4D*). Virtual co-expression analysis shows that affected *sst1.1* clusters partially overlap with *optb* in the hypothalamus and are either intermingled with or located just lateral to DA clusters (*Figure 4—figure supplement 1H*). The observation that *sst1.1* deficits are associated with loss of *otpb* suggests a potential mechanism. As previously noted, *fezf2* is required for *otp* expression in the ventral diencephalon (*Blechman et al., 2007*; *Wolf and Ryu, 2013*) and—at least in the context of the zebrafish hindbrain—*otp* is required for the development of *sst1.1* neurons (*Fernandes et al., 2013*). Although a previous publication found no *sst1.1* defects in *fezf2* mutants (*Blechman et al., 2007*), this conclusion was likely based on the strongly-expressing *sst1.1* clusters in the anterior hypothalamus, which appear normal in our analysis (*Figure 3A,C*). Changes in the highly punctate *sst1.1* expression domains may be easier to detect when using averaged 3D reconstructions from multiple embryos, as opposed to viewing individual embryos.

### *Fezf2* is required for glutamatergic development in the zebrafish telencephalon

The second phenotypic hotspot in *fezf2* mutants is the telencephalon (*Figure 3A,B*). Surprisingly, previous studies of *fezf2* phenotypes report that morphology and patterning of the telencephalon is normal during early development. In contrast to the diencephalon, the size and number of *th*-expressing DA cells in the telencephalon is normal in mutants during the first 2 weeks of development (*Guo et al., 1999*; *Levkowitz et al., 2003*; *Rink and Guo, 2004*), although TH immunoreactive processes in the subpallium are reduced at later stages (*Rink and Guo, 2004*). Similarly, telencephalic expression of the GABAergic marker *gad1b* is reported to be normal in *fezf2* deficient embryos (*Yang et al., 2012*). Consistent with these studies, our Automated Correlation Analysis shows DA markers (*th, otpb, vmat2*) in the telencephalon are essentially normal, although there may be small changes in *th* expression at 3 dpf (*Figure 3A*). In contrast, GABAergic markers (*dlx5a, gad1b*) and glutamatergic markers [*vesicular glutamate transporter 1* (*vglut1*), *vglut2.2*] show reduced correlation in the telencephalon. MIPs confirm that GABAergic and glutamatergic markers show areas of significant difference in the telencephalon, while DA markers remain unaffected (*Figure 3C*; *Supplementary file 2*). Glutamatergic neurons exhibit a striking reduction in the telencephalon at 2 and 3 dpf while the GABAergic phenotype is more complex and appears to encompass both areas of reduced and increased expression.

Detailed examination of 2D slices and 3D segmentations within the telencephalon confirms our automated phenotyping results and provides additional details about the *fezf2* phenotype. DA markers, expressed in bilateral domains in the telencephalon, are largely unaltered in *fezf2* mutants (*Figure 5A*; *Figure 5—figure supplements 1B* and *2*). Defects in GABAergic markers (*dlx5a, gad1b*) are subtle and appear to be driven by alterations in 3D shape rather than by a change in volume (*Figure 5B,E*; *Figure 5—figure supplement 2*). This impression is confirmed by quantifying 3D segmentations from *dlx5a*-stained embryos. Volume measurements show no significant difference in spite of clear changes in overall shape (*Figure 6A*). 2D sections through the pallium show that GABAergic markers exhibit a small lateral shift away from the midline ventricle in mutants at 2 and 3 dpf (*Figure 5B,E*; *Figure 5—figure supplement 1D*). *Gad1b* also appears to be slightly reduced in the posterior region of the pallium (*Figure 4B*). In contrast to the GABAergic phenotype, glutamatergic defects in the telencephalon are highly pronounced when examined in slices and 3D segmentations and clearly reflect a substantial decrease in volume (*Figure 5C,F*; *Figure 5—figure supplement 2*; *Video 3*). *Vglut1* expression is primarily restricted to the pallium, while *vglut2.2* is more broadly expressed throughout the pallium, subpallium, and olfactory bulb (*Figure 3—figure supplement 2*). 2D slices and volume measurements from 3D segmentations confirm that both markers are significantly reduced in *fezf2* mutants, with loss of the pallial glutamatergic population being particularly dramatic. In addition to deficits in *vglut* genes—which mark mature glutamatergic populations—mutants show a reduction in telencephalic *tbr1b*, a T-box transcription factor that plays an essential role in specifying glutamatergic pyramidal neurons in the mammalian neocortex (*Englund et al., 2005*; *Hevner et al., 2006*) (*Figure 5—figure supplements 1E* and *2*). Similar to *dlx5a* and *gad1b*, *tbr1b* undergoes a slight lateral shift away from the telencephalic midline ventricle.

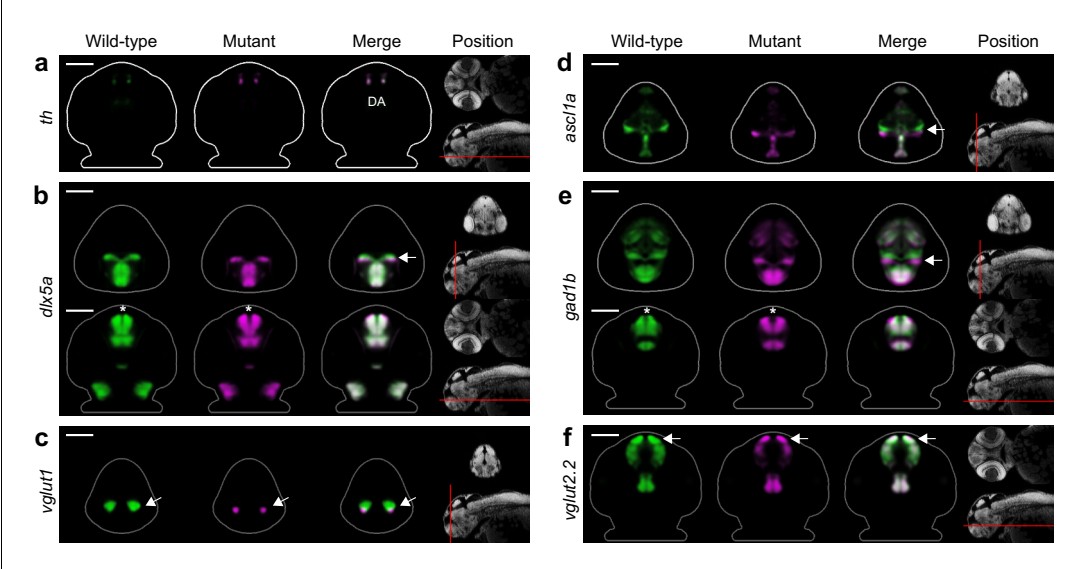

**Figure 5.** *Fezf2* mutants exhibit telencephalic glutamatergic deficits during early development. (A–F) Overlay analysis of in situ expression patterns in wild-type and *fezf2* mutant embryos at 2 dpf. Wild-type expression patterns are shown in green and *fezf2* mutants are shown in magenta. (A) Expression of *th* is largely unaffected in dopaminergic (DA) clusters of the telencephalon. (B) Expression of the GABAergic marker *dlx5a* is shifted ventrally in the diencephalon (arrow) and is away from the midline in the telencephalon (asterisk), but is not significantly reduced. (C) Expression of the glutamatergic marker *vglut1* is substantially reduced in the telencephalon (arrow). (D) Expression of the progenitor marker *ascl1a* is shifted ventrally (arrow) and compressed along the dorsoventral axis of the telencephalic midline. (E) Expression of the GABAergic marker *gad1b* is shifted ventrally in the diencephalon (arrow) and is away from the midline in the telencephalon (asterisk), but is not significantly reduced. (F) Expression of the glutamatergic marker *vglut2.2* is substantially reduced in the telencephalon (arrow). (A–F) The position and orientation of each 2D slice within the embryo is indicated on the Nissl-stained two photon reference image to the right of the panel. For frontal sections anterior is to the top; for transverse sections dorsal is to the top. Scale bar: 100 µm.

The following figure supplements are available for figure 5:

**Figure supplement 1.** Additional telencephalon phenotypes in *fezf2* mutant embryos.

**Figure supplement 2.** 3D segmentations of telencephalic expression domains.

Our findings represent the first evidence that *fezf2* is required for development of glutamatergic neurons in the zebrafish telencephalon.

In addition to GABAergic and glutamatergic defects, many other probes show reduced correlation in the telencephalon (*Figure 3A*; *Supplementary file 2*). These include markers of multipotent progenitor subpopulations (*aslc1a*, *Figure 5D*; *eomesa*, *neurog1*, *zic2a*, *Figure 5—figure supplement 1G–I*) (*Houtmeyers et al., 2013*; *Wilkinson et al., 2013*; *Englund et al., 2005*), a marker of astrocytic and glycinergic differentiation (*glyt1*, *Figure 5—figure supplement 1C*) (*Betz et al., 2006*), and a transcription factor broadly expressed within the zebrafish telencephalon (*emx3*, *Figure 5—figure supplement 1F*) (*Viktorin et al., 2009*; *Ganz et al., 2014*). As in the diencephalon, the *fezf2* expression domain itself is highly disorganized in mutants (*Figure 5—figure supplements 1A* and *2*). Closer examination reveals several common features associated with telencephalic deficits. Many markers are compressed along the dorsoventral axis of the telencephalon in mutants, a change that is sometimes accompanied by reduced volume (*Figure 5—figure supplement 2*; note reduction in *emx3* and *eomesa*, *Figure 5—figure supplement 1F,G*). Dorsoventral compression is most evident in markers with strong midline ventricle expression such as *ascl1a* (*Figure 5D*). When loss of expression occurs it is particularly pronounced in the pallium, as seen when markers broadly expressed in the pallium and subpallium (*emx3*, *tbr1b*) are co-registered with markers that are more restricted to the subpallium (*dlx5a*, *gad1b*) (*Figure 6B,C*). Lastly, many diencephalic expression domains located in close proximity to the telencephalon undergo an anterior/ventral shift. When mutant and wild-type expression patterns (labeled in magenta and green, respectively) are overlaid,

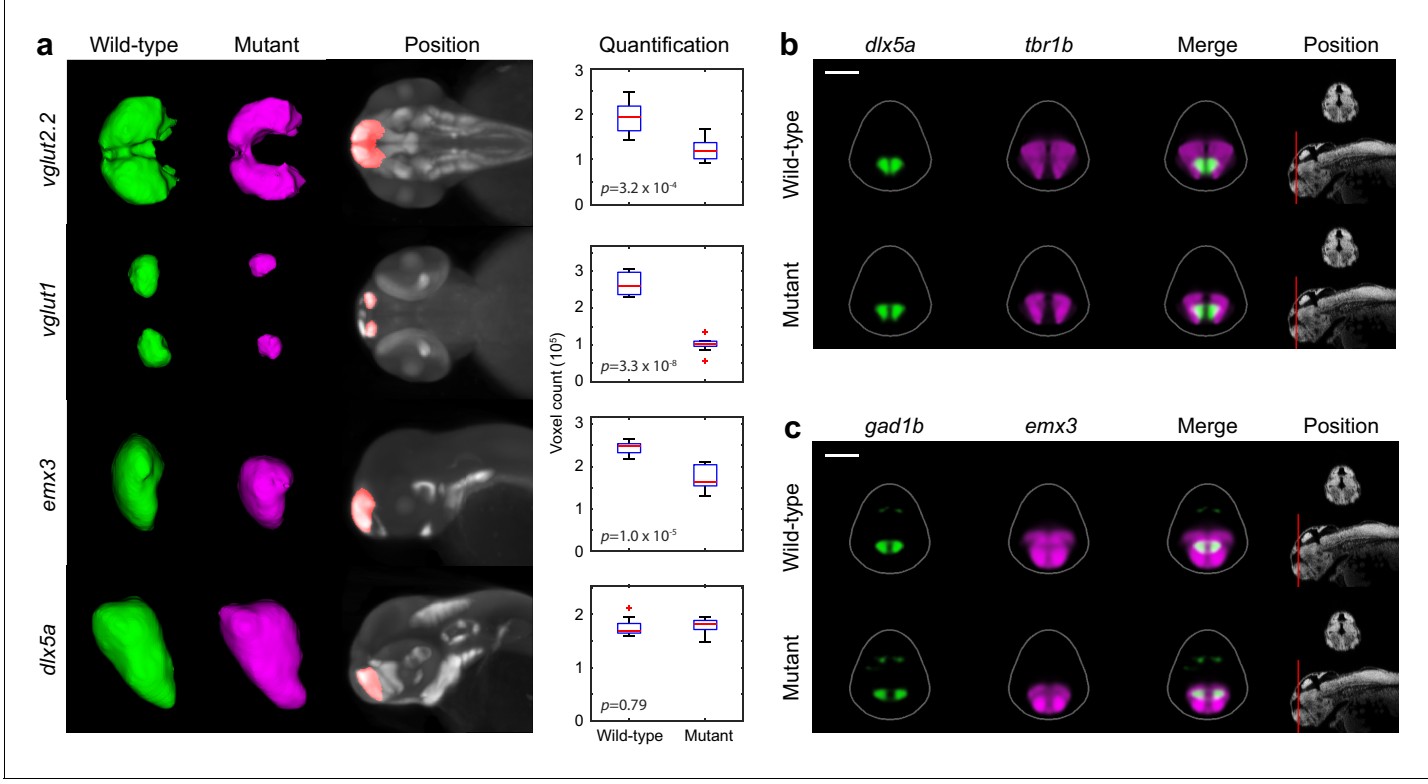

**Figure 6.** Segmentation and volume measurements. (A) 3D segmentations and volumetric quantification of gene expression domains in the telencephalon of wild-type (green) and *fezf2* mutant (magenta) embryos at 2 dpf. 3D segmentations are done independently on eight or more embryos per experimental group; averaged wild-type and mutant segmentations are shown. Position of 3D segmentations is indicated on maximum intensity projections. Box-and-whisker plots show volume measurements from 3D segmentations of the indicated probes within the telencephalon. Tops and bottoms of each box represent the 25th and 75th percentiles of the samples, respectively. Whiskers are drawn from the ends of the interquartile ranges to the furthest observations that fall within ±1.5 times the interquartile range away from the top or bottom of the box. The line in the middle of each box is the sample median. Observations beyond the whisker length are marked as outliers (+ sign). Statistical significance was determined by two-tailed *t*-test. Volume measurements used for box-and-whisker plots are available in **Figure 6—source data 1**. (B–C) Multi-probe co-expression analysis of telencephalic deficits in *fezf2* mutants. The position and orientation of each 2D slice within the embryo is indicated on the Nissl-stained two photon reference image to the right of the panel. Dorsal is to the top. Scale bar: 100 μm.

The following source data and figure supplement are available for figure 6:

**6–Source data 1.**
**Figure supplement 1.** Two-photon analysis of telencephalon morphology.

this shift shows up as a narrow band of magenta running along the anterior/ventral edge of the expression domain. This shift is evident in *gad1b* (**Figures 4B** and **5E**), *dlx5a* (**Figure 5B**), *ascl1a* (**Figure 5D**), and *neurog1* (**Figure 5—figure supplement 1H**). Taken together, these observations suggest the size of the telencephalon (particularly the pallium) may be reduced in *fezf2* mutants, resulting in a concomitant anterior/ventral shift in neighboring diencephalic domains when embryos are co-registered. To verify a physical reduction in the telencephalon, we performed 2P excitation microscopy on Nissl-stained wild-type and mutant embryos at 2 dpf. Using Nissl staining to visualize brain morphology, we generated manual 3D segmentations of the telencephalon based exclusively on physical landmarks (**Figure 6—figure supplement 1A**). Volume measurements showed a highly significant reduction in overall telencephalic size in *fezf2* mutants (**Figure 6—figure supplement 1B**). Additionally, the brain ventricle of the dorsal telencephalon is noticeably larger in mutants based on sections through Nissl-stained images (**Figure 6—figure supplement 1C**).

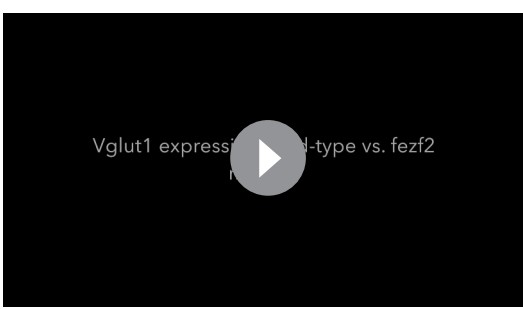

**Video 3.** Visualization of *vglut1* expression in wild-type and *fezf2* mutant embryos. The video shows overlay analysis of the *vglut1* expression pattern in wild-types and mutants at 2 dpf. Wild-types are shown in green and mutants in magenta. The position and orientation of each 2D frontal slice within the embryo is indicated on the Nissl-stained two photon reference image to the left.

## Discussion

We present a platform and algorithms for rapid 3D deep-phenotyping of zebrafish embryos using chromogenic WISH. Automated quantitative phenotyping approaches like this will be invaluable for characterizing the numerous new lines being developed using advanced genome editing techniques. 3D datasets are essential for understanding developmental patterning and for systems-level analysis of gene regulatory networks and neural circuits. Published techniques for automated 3D analysis of gene expression in zebrafish are limited to fluorescent readouts and optical sectioning approaches (*Ronneberger et al., 2012*). Although spectrally distinct fluorescent probes offer the advantage of simultaneously visualizing three or more genes per sample, they also require expensive hardware for 3D imaging, suffer from photobleaching, anisotropic resolution, have low signal intensity relative to brightfield labels, and undergo quenching under common tissue clearing protocols. Additionally, even fluorescent probes require registration to a common reference frame for simultaneous analysis of more than a few genes or for comparisons between experimental groups. Chromogenic WISH staining offers a complementary approach to 3D phenotyping. Photobleaching and quenching are not a concern, stains are extremely high-contrast (allowing for higher-throughput imaging) and stable for years, and protocols are well-established, robust, inexpensive, and have been fully automated (*Hauptmann, 2015*). Quantitative deep-phenotyping of chromogenic WISH-stained embryos requires: (1) hardware for rapidly imaging cleared non-embedded samples, (2) algorithms for aligning reconstructions to a common reference frame, and (3) robust analysis tools to correlate gene expression between experimental groups. Our automated WISH OPT platform addresses all of these requirements.

Using deep phenotyping to analyze zebrafish *fezf2* mutants, we have detected several overlooked deficits. We show for the first time that mutants exhibit defects in glutamatergic neurogenesis in the telencephalon and that this phenotype is already present at 2 dpf. These data reveal unsuspected commonalities between *fezf2* function in fish and mammals and suggest an evolutionarily conserved role in glutamatergic development, particularly in the dorsal telencephalon. Our findings help to resolve a paradox: although developmental expression of *fezf2* in the brain is highly conserved across species, reported loss- and gain-of-function phenotypes in zebrafish and rodents differ considerably. Zebrafish mutants are characterized by diencephalic deficits in DA, 5-HT, and GABAergic populations, while knockout mice display telencephalic deficits in cortical glutamatergic projection neurons. In mice, ectopic expression of *Fezf2* in neural progenitor populations that normally give rise to GABAergic neurons redirects them to a glutamatergic fate (*Rouaux and Arlotta, 2010*; *Zuccotti et al., 2014*). Consistent with this, *Fezf2* has been shown to promote glutamatergic corticospinal motor neuron identity and repress GABAergic identity in mammalian cells by activating *Vglut1* and repressing *Gad1* (*Lodato et al., 2014*). In contrast, overexpression of *fezf2* during early zebrafish development results in increased DA neurons in the ventral diencephalon (*Yang et al., 2012*). Until now, no *fezf2* glutamatergic defects have been described in zebrafish, raising the possibility that *fezf2* has assumed novel functions in the mammalian telencephalon. Based on our findings, it seems likely *fezf2* actually has a longstanding role in telencephalic glutamatergic differentiation predating the divergence of teleosts and tetraopods. Significantly, glutamatergic populations are most strongly affected in the pallium of *fezf2* mutants, the same region of the telencephalon from which the mammalian neocortex is derived.

In addition to revealing a common role for *fezf2* in glutamatergic neurogenesis, our findings bring *fezf2* defects during early zebrafish development into closer agreement with phenotypes described in adult fish. Recently it has been reported that adult *fezf2* mutants have strikingly smaller

telencephalons, although gross morphological changes are not apparent even at 2 weeks post fertilization (*Berberoglu et al., 2014*). Telencephalic reduction is accompanied by distortions in *fezf2* expression and enlargement of the dorsal telencephalon ventricle. Reduced telencephalic size, altered *fezf2* expression, and ventricular enlargement is the same constellation of phenotypes we detect at 2 dpf using deep-phenotyping, although all deficits are far less pronounced at this early stage and would be difficult to detect without quantitative 3D approaches. Previous studies have also reported decreased neurite complexity and increased numbers of neural stem cells in the dorsal telencephalon of adult *fezf2* mutants, suggesting a failure of maturation and differentiation (*Berberoglu et al., 2014*). This interpretation is supported by our finding that pluripotent progenitor markers (*ascl1a*, *ascl1b*, *neurog1*) are not significantly reduced in the *fezf2* mutant pallium, while markers of some postmitotic subtypes (*tbr1b*, *vglut1*, *vglut2.2*) undergo a substantial reduction (*Figure 5—figure supplement 2*, *Supplementary file 2*).

## Materials and methods

### Maintenance of fish, embryo collection, and genotyping

Zebrafish were maintained under standard laboratory conditions and staged as described (*Westerfield, 1995*, *Kimmel et al., 2001*). All procedures on live animals were approved by the Massachusetts Institute of Technology Committee on Animal Care. Zebrafish bearing the *fezf2^{m808}* mutation (also known as *too few; tof^{m808}*; RRID:ZFIN_ZDB-ALT-980520-28) were a generous gift of S Guo, UCSF, San Francisco, CA (*Rink and Guo, 2004*). Homozygous mutant *fezf2^{m808}* embryos and age-matched wild-type siblings were obtained by crossing heterozygous *fezf2* adults that had previously been backcrossed to the TAB14 wild-type background (Zebrafish International Resource Center catalog #ZL1438). In all experiments, 0.2 mM 1-Phenyl-2-thiourea (PTU) was added to the embryo medium to inhibit melanogenesis (*Karlsson et al., 2001*). Embryos were fixed at the indicated stages overnight at 4°C in 4% paraformaldehyde and stored in methanol at −20°C.

Embryonic and adult zebrafish were genotyped using the Derived Cleaved Amplified Polymorphic Sequences (dCAPS) method to detect the *fezf2^{m808}* mutation (*Yanagisawa et al., 2003*). PCR conditions consisted of 37 cycles of 95°C for 20 s, 60°C for 20 s, and 68°C for 30 s. The following PCR primers were used for the dCAPS assay, resulting in the introduction of an *Alu*I-sensitive restriction site in the mutant but not the wild-type *fezf2* allele (single nucleotide mismatch in the reverse primer is underlined).
Forward: GCTCTTCTGACGGGAAACCC
Reverse: TACACAACGTGCTGGCTTGTCGGAAACCTTTCC<u>A</u>GC

Paraformaldehyde-fixed embryos were genotyped by removing the tail tip using a surgical scalpel blade. Tips were transferred to 96-well plates or PCR strip tubes and PCR-ready DNA was isolated by resuspending in 10 μL nuclease-free water containing 100 μg/mL Proteinase K (Invitrogen, Carlsbad, CA), incubating at 55°C for 1–2 hr, and then incubating at 95°C for 10 min to heat inactivate the Proteinase K. PCR reactions were performed using 1 μL of tail tip lysate in a total reaction volume of 12.5 μL.

### Probe synthesis and WISH staining

We generated a custom in situ probe library by using RT-PCR to amplify marker genes of interest from total RNA isolated and pooled from zebrafish embryos between 1–6 dpf (see *Supplementary file 1* for genes, probe lengths, and RT-PCR primer pairs). PCR fragments were ligated into a pCRII vector using the TOPO TA Cloning Kit (Invitrogen). Digoxigenin-labeled RNA probes were transcribed from linearized template using a SP6/T7 DIG RNA Labeling Kit (Roche, Basel, Switzerland) and purified using the RNeasy Mini Kit RNA cleanup protocol (Qiagen, Hilden, Germany).

Whole-mount in situ hybridization (WISH) was performed essentially as described (*Thisse and Thisse, 2008*), but with several important modifications that minimize physical distortions to fixed embryos during high temperature hybridization steps, optimize the robustness of the staining reaction, and increase the stability of stained samples during clearing and tomographic imaging steps. All modifications from the standard zebrafish WISH protocol (ZFIN Protocol Wiki, Thisse Lab 2010 update) are included in the summary below:

### WISH day 1

1. Setup: for each probe, wild-type and mutant embryos should be kept in the same vial, tube, or basket for all WISH steps in order to minimize the possibility of artifacts arising from differences in sample handling and processing. In order to visually distinguish wild-type and mutant embryos after PCR genotyping, small but distinctive additional cuts can be made to the tail tips of one class.
2. Permeabilization: rehydration is carried out as described and embryos are permeabilized using 10 µg/mL of Proteinase K (Invitrogen) in PBST. The time of treatment varies with the age of the embryo and should be determined experimentally for each lot of Proteinase K. We used 5 min and 10 min at room temperature for 2 dpf and 3 dpf embryos respectively.
3. Postfixation: following permeabilization, embryos are washed briefly 2 times in PBS + 0.1% Tween 20 (PBST) to remove the Proteinase K. They are then postfixed in a solution of 4% paraformaldehyde and 0.2% EM grade glutaraldehyde (Sigma-Aldrich, St. Louis, MO) in PBST for 30 min at room temperature and washed 3 times (5 min per wash) in PBST.
4. Prehybridization: embryos are transferred to baskets with nylon mesh bottoms to facilitate subsequent media changes. Baskets are then moved into 15 mL tubes containing room temperature hybridization mix (HM; 50% Deionized Formamide, 5X SSC, 50 µg/mL Heparin, 0.1% Tween 20, 10 mM Citric Acid, 500 µg/mL RNase-free tRNA, pH 6.0). Prehybridization is carried out for 5 hr by placing the tubes in a water bath at room temperature and adjusting the temperature to 65°C. This allows the HM Solution to gradually warm to 65°C and appears to reduce bending in the trunks and tails of fixed embryos.
5. Hybridization: following prehybridization, embryos are transferred to HM solution containing 1–3 ng/µL of dig-labeled riboprobe (as quantified using a NanoDrop Spectrophotometer or equivalent) that has been pre-heated to 65°C. Incubation is done overnight at 65°C.

### WISH day 2

1. Post-hybridization washes: post-hybridization washes are carried out as described with the exception that the incubation temperature is reduced to 65°C for all heated steps and embryos are allowed to slowly cool to room temperature (~15 min) between the final 65°C wash step and the first room temperature wash step.
2. Antibody incubation: embryos are incubated for 4 hr at room temperature in blocking buffer [PBST containing 2% Sheep Serum (vol/vol) (Sigma-Aldrich) and 2 mg/mL BSA (Sigma-Aldrich)]. Simultaneously, the Anti-Digoxigenin-AP, Fab fragments (Roche; RRID:AB_514497) are diluted 1:5000 in blocking buffer and pre-absorbed for 4 hr at room temperature. Embryos are then transferred to the pre-absorbed Anti-Digoxigenin-AP antibody and incubated overnight at 4°C.

### WISH day 3

1. Washes and labeling: washes and chromogenic labeling steps are done essentially as described. The labeling mix is made by adding 15 µL of NBT/BCIP Stock Solution (Roche) to 1 mL alkaline Tris buffer (100 mM Tris HCl pH 9.5, 50 mM MgCl$_2$, 100 mM NaCl, 0.1% Tween 20) and is always prepared fresh and shielded from light prior to use. Staining is carried out at room temperature in the dark and progress of the staining reaction is monitored periodically. Optimal staining times vary depending on the probe, the developmental stage, and the degree of sample permeabilization. If staining remains weak after several hours, embryos may be transferred to 4°C and the staining reaction can proceed overnight.
2. Post-staining: To stop the chromogenic staining reaction, embryos are washed twice in PBST and then three times in stop solution (PBST, 1 mM EDTA) for 15 min per wash. Embryos are then post-fixed in 4% paraformaldehyde at room temperature for 1 hr. For long-term storage embryos are dehydrated in a stepwise fashion into methanol.

### Counterstaining and clearing

Prior to tomographic imaging, embryos are counterstained using SYTOX Green (Thermo Fisher Scientific, Waltham, MA), which imparts a faint red color to the body that can be visualized in the blue or green channels and aids in tomographic reconstruction (see 'image reconstruction' below). Embryos are also cleared using benzyl alcohol and benzyl benzoate (BABB). Counterstaining and clearing are done using the following protocol:

- Rehydrate stained embryos by successive 5 min washes at room temperature in the following:

- - 25% PBST +75% methanol
  - 50% PBST +50% methanol
  - 75% PBST +25% methanol
- Wash two times, 5 min per wash, in PBST
- Dilute SYTOX Green nucleic acid stain 1:500 in PBST and incubate embryos at 4°C overnight.
- Dehydrate embryos by successive 5 min washes at room temperature in the following:
  - 75% PBST +25% methanol
  - 50% PBST +50% methanol
  - 25% PBST +75% methanol o 100% methanol
- Wash embryos three times in methanol (10 min per wash).
- Remove methanol and wash twice in BABB [5 parts benzyl benzoate (Sigma-Aldrich) and 2 parts benzyl alcohol (Sigma-Aldrich)].

## High-throughput optical projection tomography platform

The configuration of all components of the OPT platform is illustrated in *Figure 1A*. A broadband Tungsten-Halogen lamp (#QTH10; Thorlabs, Newton, NJ) is used for illumination. The light from the lamp passes through a 5 mm thick ground glass diffuser before entering the imaging chamber containing the WISH-stained sample and is then collected by a telephoto zoom lens (EF 70–300 mm f/4–5.6 L IS USM; Canon, Tokyo, Japan). Alternatively, a 4X microscope objective (along with a tube lens and 5 mm aperture) may be used in place of the telephoto lens. We have tested a Plan Fluor 4X/0.13 NA objective (Nikon, Tokyo, Japan) and obtain 3D reconstructions of comparable quality and resolution to those produced by a telephoto lens. A high-speed CCD camera (Prosilica GX 1050C; Allied Vision Technologies, Stadtroda, Germany) is used for acquisition. The imaging chamber includes a stepping motor (AH1K-S543, Autonics, Busan, South Korea) run by 5-phase micro-stepping drivers (MD5-HF14, Autonics), which holds and rotates the aluminosilicate glass capillary (#A150-100-10, ID = 1 mm; Sutter Instrument, Novato, CA) during image acquisition. The lower part of the capillary that holds the embryo is immersed in BABB, which is contained within a transparent glass cuvette (#C-G-F-20, The Science Outlet, Keota, OK), to ensure refractive index (RI) matching between the inside and outside of the capillary. All components are mounted to an optical table or breadboard and the light source, imaging chamber, lens, and camera are aligned. To ensure precise positioning of embryos of different sizes from various developmental stages and to increase embryo stability during rotational imaging, a tapered insertion is synthesized from index-matching optical adhesive at the bottom end of the capillary (*Figure 1B*; details below). Individual zebrafish embryos are loaded through the single upper fluidic port, enter the vertically configured glass capillary, and are stopped when they reach the RI-matched tapered insertion, which is precisely aligned with the telephoto zoom lens. The stepping motor then automatically rotates the capillary through a complete revolution in ~12 s as images are acquired. Following image acquisition the sample is unloaded through the same upper fluidic port and the next sample is introduced. Loading and unloading is achieved by pumping or withdrawing BABB from the fluidic port using a polypropylene syringe. Average handling time per embryo is 55.0 ± 2.9 s when loading and unloading is done manually (embryo loading: 19.4 ± 2.1s; image acquisition and data transfer: 18.0 ± 0.7s; embryo unloading: 17.6 ± 3.0).

## Fabrication of index-matching capillary insertions

The aluminosilicate glass capillary is first cleaned by immersion in NanoStrip (KMG, Pueblo, CO) for 30 min and then thoroughly rinsed in DI water. Once dry, the distal end of the capillary is dipped into index-matching optical adhesive (#NOA 61; Norland Products, Cranbury, NJ) and held in place until the capillary draws up ~5 mm of the liquid (*Figure 1—figure supplement 1A*). The tapering is then formed through a two-step UV curing procedure. In the first step, the capillary filled with uncured adhesive is rotated using a miniature DC electric motor (#EL292-0015S, Ajax Scientific, Scarborough ON) under a 254 nm short wavelength UV source (Spectrolinker XL-1000, Spectroline, Westbury, NY) for ~240 s (*Figure 1—figure supplement 1B*). During this step, the capillary is positioned ~15 cm from the light source, resulting in an estimated irradiance of ~6 mW/cm$^2$ at the surface. Because the UV 254 wavelength has a poor optical transmission rate in the adhesive, it is only cured in a thin cylindrical outer shell immediately adjacent to the capillary wall. An opaque mask is used to shield the proximal half of the adhesive from the UV source during the first curing

step. In the second step, air is pumped through the capillary from the proximal end using a 60 mL syringe at a rate of 20 mL per second, causing the uncured adhesive to form a tapering surface (*Figure 1—figure supplement 1C*). As air is being passed through the capillary, a 365 nm long wavelength 100-watt UV lamp (Blak-Ray B-100A, UVP, Upland, CA) is used to rapidly cure the adhesive into the desired shape. During this step, the capillary is positioned ~10 cm from the UV lamp, resulting in an estimated irradiance of ~10 mW/cm$^2$ at the surface. Once the adhesive has begun to solidify, the distal end of the capillary is connected to a vacuum line and allowed to cure for an additional 10 min under long wavelength UV. Finally, the capillary is incubated at 50°C for 12 hr to enhance the solvent-resistance of the adhesive.

## Image acquisition

During image acquisition, a custom-made MATLAB program controls the stepping motor and CCD camera as the capillary rotates through 360° and 360 images of the embryo are continuously acquired at a frame rate of 30 fps. Each image is 1024 × 1024 pixels (~1540 × 1540 µm field of view) and provides sufficient resolution to visualize WISH stains that label individual cells (e.g. *th*, which labels discrete cells in the retina). CCD cameras often have minor variation in the pixel response, which can result in image artifacts (*Figure 1—figure supplement 1D*). To correct for this, we illuminated the camera using the OPT light source with the imaging chamber removed from the light path to achieve uniform illumination and calculated the average pixel response for each pixel over time. Dividing the average individual pixel response by the average value for all pixels provides an estimation of the deviation from the average. For all acquired images, each pixel is divided by its estimated deviation value, thereby eliminating artifacts arising from pixel sensitivity variation (*Figure 1—figure supplement 1E*) (*Walls et al., 2005*).

## Image reconstruction

The orientation of the embryo in the capillary (head-first vs. tail-first) is automatically detected as described (*Chang et al., 2012*). Movements of the capillary that occur during rotation and image acquisition need to be corrected to ensure high-resolution reconstructions. All frames are aligned based on methods we developed previously for non-embedded OPT imaging (*Pardo-Martin et al., 2013*). We correct for sideways movements of the capillary by detecting the sharp boundaries at the edges of the capillary. The upper and lower boundaries are used to align all rotational images and to stretch or shrink the image to achieve an equal capillary diameter across all images. The addition of a tapered capillary insertion, in combination with the vertical orientation of our imaging chamber, eliminates artifacts arising from longitudinal movements of the embryo along the capillary axis. The precise center of rotation (COR) is determined by generating multiple 2D reconstructions through the embryos using different estimated CORs. The COR whose reconstruction has the lowest entropy (i.e. the sharpest boundaries) is then used for the subsequent 3D reconstruction of the entire embryo (*Pardo-Martin et al., 2013*). Correction for non-uniform illumination across the cross-section of the capillary is done by first determining the maximum value for all pixels in all rotational frames. Regions containing the embryo are then masked out and an interpolation is done in the masked out region to create a blank background image. This background image is used as a baseline to equalize the illumination. After the alignment, all color channels are independently reconstructed with a GPU implemented Filtered Back Projection algorithm provided in the Astra toolbox (*van Aarle et al., 2015*). Each of the three color channels contains slightly different information. The dark blue chromogenic in situ stain is most easily distinguished from the background in the red channel. In the blue channel the in situ stain is faint and the SYTOX Green counterstain dominates, while in the green channel both SYTOX and in situ stains can be clearly seen *Figure 1—figure supplement 1F–I*). The alignment and reconstruction takes ~11 min per embryo to run on a 2.6 GHz hexa-core processor standard workstation and the reconstructed image has a size of 512x512 × 512 voxels. Both alignment and reconstruction are fully automated and can therefore be run offline without any user intervention.

## Image registration

All registrations discussed below are done with the toolbox elastix (RRID:SCR_009619) (*Klein et al., 2010*; *Shamonin et al., 2013*) and MATLAB Image Processing Toolbox (Release R2015b;

MathWorks, Natick, MA). There are four types of transforms used for registration in this paper: rigid, similarity, affine, and B-spline. A rigid transform consists of only translation and rotation. The similarity transform consists of translation, rotation, and isotropic scaling. The affine transformation consists of translation, rotation, scaling, and shearing. The B-spline transform is a deformable transform where a deformation field is modelled using B-splines. The deformation is achieved by moving the control points in a sparse regular grid. The deformation at any point is obtained by using a B-spline interpolation kernel. A regularizer is used to constrain the deformation of the B-spline from deforming too much. We use a local rigidity penalty term developed for medical image registration in all deformable registrations (*Staring et al., 2007*).

The measure of similarity used in all registrations is normalized correlation, which is computed using elastix (*Klein et al., 2010*; *Shamonin et al., 2013*). Furthermore, all registrations are done on gradient magnitude images to emphasize edges. In addition, all registrations are done with a multi-resolution approach (*Lester and Arridge, 1999*). The registration starts with a low resolution image (downsampled 6× or 8×), providing a coarse registration. The registration continues progressively through to higher resolutions. The final registration is done on a 2× downsampled or full-size image. This registration scheme makes the optimization less likely to get stuck in a local minimum and also decreases the time to reach the global minima.

Our basic registration workflow consists of three steps: initial, coarse, and fine alignment. The initial alignment is a rigid registration with the aim of positioning all embryos in a roughly similar position to the reference image. Eight different initial positions are tested for the moving image. This step is done on an 8× downsampled image. The position that provides the best match based on the similarity measure is further aligned in the next step. After the initial alignment, we perform a coarse registration with a similarity and/or an affine transform. This step is done with an 8×, 4×, and 2× downsampled image. This provides a good registration without adding any deformation to the sample. The fine registration is the last step of the process (and is omitted in certain instances as noted) and consists of a deformable registration. This step is done an 8×, 6×, and 3× downsampled image and performs a non-linear deformation to the sample. We used a grid spacing of 60 voxels in order to limit the deformation to larger deformations.

## Unstained reference fish

Before individual reconstructions of WISH-stained embryos can be compared, they must first be brought into alignment through registration to an appropriate reference embryo that is compatible with all probes in the library. In order to generate a suitable reference, we first perform tomographic reconstructions on 5–7 wild-type fish that have been taken through the WISH protocol without a riboprobe, ensuring that registration will be driven by morphology rather than features unique to any given gene expression pattern. An average unstained reference fish (URF) is then created using the registration workflow outlined above in combination with Iterative Shape Averaging (ISA) (*Rohlfing et al., 2001*). The initial alignment of all unstained embryos is done using a rigid transform. An average fish is then created from the aligned images. In the next round of registration, all embryos are aligned to the average using a similarity transform. A new average is then created from the aligned images and the process is repeated with an affine transform and finally a deformable transform to produce the final average URF (*Figure 1—figure supplement 2A*). An age-specific URF is created for each developmental stage of interest (i.e. 2 and 3 dpf in the present paper). All registration steps for the URF are done on the green channel.

## In situ pattern alignment

When viewed in bright field (blue or green channels), SYTOX Green labels all tissues with a more or less uniform faint red counterstain. This has the effect of highlighting surface features and a few prominent morphological landmarks (such as brain ventricles), but does not delineate internal brain structure sufficiently to allow for precise deformable registration of internal points. In contrast, the gene expression patterns visible in in situ stained embryos (red or green channels) provide prominent internal reference points to guide deformable registration. Indeed, the use of deformable transforms is essential to precisely align complex 3D expression patterns and minimize the possibility of incorrectly detecting alterations where none exist during subsequent automated statistical analysis. To overcome these challenges and align WISH-stained embryos as closely as possible to one another

and to the common reference frame, we begin by a selecting a probe-specific reference for each probe in the library. Probe reference fish (PRFs) are chosen using a variation of the strategy used to create URFs. All 8+ individual wild-type embryos for a given probe at a given developmental stage are first registered together as previously described using ISA in the green channel (*Figure 1—figure supplement 3A*). At the end of this process, the individual embryo that is most similar to the final average wild-type (based on the correlation value from the last iteration of ISA) is chosen as the PRF (*Figure 1—figure supplement 3B*). The goal of this process is to select the embryo with the most representative overall morphology and staining pattern as the reference image, rather than to randomly select an embryo that might represent an outlier. The selected PRF is then aligned to URF with a rigid and affine registration in the blue channel (*Figure 1—figure supplements 2B* and *3B*). The blue channel is used in order to minimize the influence of the in situ pattern on the registration to the unstained reference. All wild-type and mutant embryos stained with the same probe are then aligned to the PRF. This is done in the green channel to make use of probe-specific landmarks and is performed using our standard registration workflow: rigid, similarity, affine and finally deformable registration (*Figure 1—figure supplements 2C* and *3C*). During alignment, the relative volume of all embryos is estimated from the affine registration to the PRF (which corresponds to the size of the URF). Occasionally, an embryo is detected that differs substantially in size from both wild-type and mutant age-matched clutchmates. Typically, these outliers are smaller in volume, suggesting sporadic developmental defects linked to embryo health or genetic background. Therefore, to avoid phenotyping artifacts, embryos that deviate by more than 50% from the volume of the URF are excluded from further analysis. Once all embryos have been registered with each other, a new average wild-type is created by calculating the average intensity of each aligned voxel in the wild-type group. This average is aligned to the URF in the blue channel with an affine transform. The transformation from this final step is then applied to all wild-type and mutant embryos for that probe, ensuring they are aligned as closely as possible to the common reference frame. Alignment takes 11 min per sample on a 2.6 GHz hexa-core processor standard workstation.

## Brain atlas

In order to create a custom 3D anatomical reference atlas for mapping brain-specific WISH staining patterns, we imaged Nissl-stained zebrafish embryos at 2 and 3 dpf using two-photon (2P) excitation microscopy. To ensure the best possible match between 3D atlases and OPT reconstructions, embryos were processed using our standard WISH protocol (without a riboprobe). Following WISH, embryos were stained with NeuroTrace 500/525 Green Fluorescent Nissl Stain (Thermo Fisher Scientific) diluted 1:100 in PBS. Staining was carried out for 10 hr at 4°C and embryos were rinsed several times in PBS. To provide a uniform counterstain and enable registration of 2P images with URFs, the embryos were further stained with SYTO 64 Red Fluorescent Nucleic Acid Stain (Thermo Fisher Scientific) diluted 1:2000 in PBS. Staining was carried out overnight at 4°C, and embryos were washed twice in PBS. Stained embryos were embedded in 0.7 mm thick disk-shaped 1% low gelling temperature agarose blocks (Sigma-Aldrich). The embedded samples were thoroughly dehydrated in methanol and then immersed in BABB for 15 min. The agarose disks were sandwiched between two thin glass slides during 2P imaging. The images were taken using a 2P workstation with Zeiss 20 × 1.0 NA DIC (UV) VIS-R objective. The excitation wavelength was set to 900 nm, the pixel resolution was set to 1.169 μm, and the step size along z-axis was 1 μm.

Manual segmentation of the brain regions was done using the software TurtleSeg 1.3 (Oxipita, Vancouver, Canada; RRID:SCR_002605) (*Top et al., 2011*, *Top et al., 2010*). Anatomical segmentation was done manually following established zebrafish brain nomenclature and with the aid of the Atlas of Early Zebrafish Brain Development (*Mueller and Wullimann, 2015*) and the ViBE-Z 3D atlas (RRID:SCR_005895; http://vibez.informatik.uni-freiburg.de/) (*Ronneberger et al., 2012*; *Rath et al., 2012*). In order to align our anatomical atlases (which based on 2P imaging) with our OPT reference frame, the SYTO red fluorescent image was aligned to the age-appropriate URF in the green channel with a rigid followed by an affine transform. Our 2 and 3 dpf 3D anatomical brain atlases comprise 22 different brain regions located within the telencephalon, diencephalon, midbrain, and hindbrain (*Figure 3—figure supplement 1*).

## Automated detection and quantification of alterations in gene expression

Automated Correlation Analysis and other automated phenotyping steps (*Figure 3A–C*) are carried out on results from the final deformable registration step. Analysis is done in MATLAB on 2× down-sampled data from the red channel, where the WISH signal is strongest. For Automated Correlation Analysis, a Pearson correlation coefficient is calculated for every probe within all brain regions based on the intensities of corresponding voxels between each wild-type and mutant embryo (*Goshtasby, 2012*). The Mann-Whitney *U* test is used to assess if intra-group correlations (wild-type vs. wild-type and mutant vs. mutant) are significantly larger than inter-group (wild-type vs. mutant) correlations ($p < 1 \times 10^{-5}$). The effect size (Cohen's *d*) is also calculated to measure the magnitude of the difference (*Sullivan and Feinn, 2012*). Small effect sizes indicate minor changes that may be due to differences in background staining or biological variation, while larger values ($d > 1$) indicate true alterations in gene expression patterns. Both *p*-value and Cohen's *d* are combined to create an Automated Correlation Analysis table depicting changes in all probes across all brain regions at 2 and 3 dpf (*Figure 3A*, left).

The voxelwise p-value test, shown together with the maximum intensity projection (MIP) in *Figure 3C* and *Supplementary file 2*, are generated by performing a Mann-Whitney *U* test to compare corresponding voxels in wild-types and mutants for each probe. A voxel is considered to be significantly increased or decreased if the *p*-value from the Mann-Whitney *U* is below $0.5 \times 10^{-3}$. The Mann-Whitney *U* test is only performed within brain regions that show alterations in the Automated Correlation Analysis for that particular probe. For all significant voxels the average wildtype is subtracted from the average mutant. Separate color coded MIPs are generated for all areas of positive difference (significant decrease in expression; cyan) and negative difference (significant increase in expression; magenta) and added to a composite MIP of the average expression pattern for the probe in both wild-types and mutants.

## Virtual 2D slicing and 3D segmentation

Because the first goal of automated phenotyping is to make an unbiased determination as to whether or not actual differences exist between wild-type and mutant embryos, it is essential to begin any analysis by treating both groups identically. This is why both wild-types and mutants are initially registered to a single wild-type PRF during Automated Correlation Analysis and other automated phenotyping steps (*Figure 3A–C* and *Figure 1—figure supplement 3C*). Using a common reference fish for both groups ensures that detected differences are due to actual alterations in gene expression rather than registration artifacts arising from the use of separate mutant and wild-type reference fish in the context of deformable (non-linear) transforms. Once it has been verified that significant differences exist, it becomes desirable to modify the registration workflow slightly prior to detailed 2D and 3D analysis, such that the final deformable step is done using separate references for mutants and wild-types. This is necessary to prevent the deformable registration algorithm from artificially minimizing actual phenotypic differences by deforming the mutant expression pattern to match the wild-type. We carry out all linear transforms (rigid, similarity, and affine) as described, and then generate an average wild-type and an average mutant embryo from the final affine step for each probe. The final non-linear B-spline transform is then done by registering all wild-types to the average wild-type reference and all mutants to the average mutant reference (*Figure 1—figure supplement 4*). This ensures that morphological differences will be better preserved during subsequent analysis. For 2D slice visualization, we use custom made VTK (*Schroeder et al., 2004*) scripts to section through 3D reconstructions from the red channel and present the results in false color (green and magenta for two color overlays; cyan, magenta, and yellow for three color overlays). The 2P Nissl stain from the aligned 3D brain atlas is used to show the slice position.

3D segmentation of gene expression patterns is implemented using adaptive thresholding on a background-reduced image. The background stain is estimated by eroding the reconstructed 3D image in the red channel, and the result is subtracted from the original reconstruction to get an image that mostly shows only the in situ signal. We then apply adaptive thresholding to segment the signal. The threshold is set using the larger value of (1) the global average plus three times the standard deviation of the pixel values of the background-reduced image, and (2) 2/3 of the maximum intensity value in the 10 μm diameter neighborhood. Results are examined and the threshold is fine-

tuned if necessary to achieve an optimal segmentation. Any obvious segmentation artifacts arising from background noise are removed manually. In order to visualize an average segmentation from features that have been independently segmented in multiple embryos (*Figure 6A*, *Figure 6—figure supplement 1A*) we use an algorithm that compares collections of segmentations and computes a probabilistic estimate of the true segmentation (*Warfield et al., 2004*). In other cases, 3D segmentations are created directly from averaged tomographic reconstructions (*Figure 5—figure supplement 2*).

## Source code

Core source code for image reconstruction and registration (https://github.com/aallalou/OPT-InSitu-Toolbox; copy archived at https://github.com/elifesciences-publications/OPT-InSitu-Toolbox) (*Allalou, 2017*), along with brain atlases (Source code 1, see below) and sample data files (Source code 2, see below), are available online. The major part of the code is written in MATLAB. The open source registration toolbox elastix (*Klein et al., 2010*; *Shamonin et al., 2013*) is used for registration and must be installed before running the code. Other required toolboxes that need to be downloaded and installed are the ASTRA Tomography Toolbox (*van Aarle et al., 2015*, *Palenstijn et al., 2011*) and DIP*image* (*Luengo Hendriks et al., 2000*). Source code for both OPT reconstruction and registration is available at https://github.com/aallalou/OPT-InSitu-Toolbox (*Allalou, 2017*). The following source code files are available for download from the Dryad Digital Repository: 10.5061/dryad.7j12m (*Allalou et al., 2017*).

**Source code 1: 3D anatomical brain atlases**. This RAR archive contains 3D brain atlases for 2 and 3 dpf embryos.

**Source code 2: Test datasets for OPT reconstruction and registration**. These RAR archive files contain the following: (1) acquisition data from our OPT platform of a 2 dpf *fezf2* mutant embryo stained with an in situ probe for *ascl1a* (folder 'data\OPT', extension '.mat'). This file serves as a test dataset for our OPT reconstruction source code. (2) Unstained reference fish (URFs; folder 'data\Registration\ referenceFish') and OPT reconstructions from 8 wild-type embryos (folder 'data\Registration\TestData_th_2dpf\wt') and 8 *fezf2* mutant embryos (folder 'data\Registration\TestData_th_2dpf\mt') stained with an in situ probe for *tyrosine hydroxylase* (*th*). URFs for 2 dpf and 3 dpf are provided. All *th*-stained embryos are 2 dpf. These images serve as a test dataset for our registration source code. Download all RAR archive files (Source code 2, parts 1 through 7) prior to extraction.

## Acknowledgements

We thank the following funding sources: NIH Director's Pioneer Award (DP1-NS082101), Packard Award in Science and Engineering, Harvard/MIT Broad Institute's SPARC Award, and a postdoctoral fellowship from the Epilepsy Foundation. All zebrafish used in these studies were raised and maintained in the Koch Institute Zebrafish Core Facility directed by Drs. Jacqueline A Lees and Adam Amsterdam. A number of the WISH probes used in this study were cloned by Pierre S Phabmixay, a research intern. We thank Dr. Su Guo and members of the Guo lab for the generous gift of the *fezf2*$^{m808}$ mutant line and for insightful discussions. We received helpful advice on optimization of the WISH protocol from a number of sources including Drs. Randall T Peterson and Colleen A Brady (Massachusetts General Hospital) and Bernard Thisse (University of Virginia). We thank Drs. Florian Engert, Alexander F Schier, and Owen Randlett (Harvard University) for advice and discussions on developing 3D anatomical brain atlases and image registration.

## Additional information

### Funding

| Funder | Grant reference number | Author |
|---|---|---|
| National Institutes of Health | Director's Pioneer Award DP1-NS082102 | Mehmet Fatih Yanik |
| David and Lucile Packard Foundation | Packard Award in Science and Engineering | Mehmet Fatih Yanik |

| The Eli and Edythe L. Broad Institute of MIT and Harvard | SPARC Award | Mehmet Fatih Yanik |
|---|---|---|
| Epilepsy Foundation | Fellowship | Amin Allalou |

The funders had no role in study design, data collection and interpretation, or the decision to submit the work for publication.

## Author contributions

AA, YW, Data curation, Software, Formal analysis, Investigation, Visualization, Methodology, Writing—original draft, Writing—review and editing; MG-R, Methodology, Writing—review and editing; PME, Conceptualization, Formal analysis, Supervision, Investigation, Writing—original draft, Project administration, Writing—review and editing; MFY, Conceptualization, Resources, Supervision, Funding acquisition, Project administration, Writing—review and editing

## Author ORCIDs

Amin Allalou, http://orcid.org/0000-0003-4028-8443
Yuelong Wu, http://orcid.org/0000-0002-4417-8236
Peter M Eimon, http://orcid.org/0000-0003-0447-517X
Mehmet Fatih Yanik, http://orcid.org/0000-0002-8963-2893

## Ethics

Animal experimentation: All procedures on live animals were performed in accordance with the recommendations in the Guide for the Care and Use of Laboratory Animals of the National Institutes of Health. Protocols were approved by the Massachusetts Institute of Technology Committee on Animal Care (protocol #0312-025-15).

# Additional files

## Supplementary files

• Supplementary file 1. Whole mount in situ probe library. All probes used for automated 3D phenotyping are listed along with the forward and reverse primers used in PCR cloning.

• Supplementary file 2. Significance analysis overlaid on maximum Intensity projections (MIPs). MIPs have been color coded to highlight all regions in which the expression of a given probe is significantly reduced (cyan) or increased (magenta) in mutant embryos ($p < 0.5 \times 10^{-3}$). Significant intensity differences between mutants and wild-types were determined by performing a Mann-Whitney $U$-test to compare corresponding voxels for each probe within each brain region showing alterations in Automated Correlation Analysis. All MIPs show dorsal (left) and lateral (right) views.

## Major datasets

The following dataset was generated:

| Author(s) | Year | Dataset title | Dataset URL | Database, license, and accessibility information |
|---|---|---|---|---|
| Allalou A, Wu Y, Ghannad-Rezaie M, Eimon P, Yanik M | 2017 | Data from: Automated deep-phenotyping of the vertebrate brain | http://dx.doi.org/10.5061/dryad.7j12m | Available at Dryad Digital Repository under a CC0 Public Domain Dedication |

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
