## [Decision Letter]

Thank you for submitting your article "Automated deep-phenotyping of the vertebrate brain" for consideration by *eLife*. Your article has been reviewed by three peer reviewers, and the evaluation has been overseen by a Reviewing Editor and Marianne Bronner as the Senior Editor.

The reviewers have discussed the reviews with one another and the Reviewing Editor has drafted this decision to help you prepare a revised submission.

Summary:

In this manuscript, Allalou et al. describe a platform that they have developed for high throughput characterisation of gene expression patterns and how they are altered in mutant zebrafish embryos and larvae. They have optimised a chromogenic in situ hybridisation protocol for optical projection tomography and developed a pipeline to register mutant and wild-type volumetric datasets to common reference brains, with anatomical annotations. The method is rigorously validated using the *fezf* mutant, reproducing both previously reported defects, anticipated changes that were previously overlooked (such as the loss of *sst1.1* within the *otp* domain) and novel defects due to a change in telencephalon size. Such changes would be difficult to identify using traditional methods for mutant characterization. Methods are for the most part well described, especially the detailed protocol describing modifications to WISH. This is a powerful new method for quantitative phenotypic characterization.

Essential revisions:

1) Although the statistical technique used for automated correlation analysis (Figure 3) is robust, the voxel-wise procedure shown in Figure 3 is not very rigorous. This is not a deal-breaker because the procedure is in any case only applied within regions already found positive using the Pearson+Whitney method. Nevertheless, strictly, the voxel-wise method used is really a sort of cluster-based technique, using arbitrary thresholds for voxel-wise significance, and cluster-size threshold. Similar methods are known to be dangerously prone to false positives (see Eklund PNAS 2016). More statistically rigorous voxel-wise analysis should be presented.

2) In several steps of the analysis, a large number of statistical tests are required to compare voxel data for numerous genes and brain regions. What was the basis for setting p value thresholds for so many comparisons?

3) Please provide more detail in support of the statement "The ability to rapidly image and align thousands of WISH-stained embryos in 3D […]". Although timings for some individual steps appears in the Methods, it would be useful to summarise in the results text the rate at which lines can be processed, imaged and registered.

4) The authors state that the registration algorithms used in other studies (e.g. CMTK, used in Randlett, 2016) are unsuitable for their WISH data. It would be informative if they could briefly comment on why this is the case.

Please make sure to submit the code and brain atlases so that they are available for download by the readers.

Also, please consider whether you would like your article to appear as a 'Tools and Resources' article or a regular one.

Suggested revisions:

The authors should offer more detailed rationale for several steps in the registration procedure; additional information in the manuscript will help inform future work by other groups. Two specific questions:

1) For each probe, one embryo (the PRF) is selected via strongest correlation to the probe-specific iterative shape average. This embryo is then registered to the URF using a rigid+affine registration (no elastic step) for the sytox channel, then other embryos registered to the PRF using deformable registration using the green channel. But with this procedure, any inaccuracy in the PRF alignment will be transmitted to all other embryos for that probe. Because initial PRF alignment is rigid+affine and not elastic, accuracy will be limited. So why is elastic registration not used for the PRF? The final registration step (subsection “Image registration”, second paragraph) is not well explained – is this rigid+affine only? How is the average wildtype created (ISA?)

2) Using the WISH pattern to drive registration to a selected wildtype embryo means all the resulting images will have low variability, but actually mask real biological variability in individual WISH patterns. The authors acknowledge this concern at the end of the manuscript – because mutant patterns are elastically aligned to a wildtype WISH pattern differences in the mutant are minimized. My understanding is that the alternate procedure in Figure 1—figure supplement 4, that avoids this problem, is only for the 2D visualization overlays in Figure 4–Figure 5, because it is obviously not suitable for statistical comparison. The more obvious procedure would be to independently align each wildtype and mutant embryo to the URF using its sytox channel, apply the transformation to the WISH channel and then perform statistical comparisons. Why was this method not used?

---

## [Author Response]

*Essential revisions:*

*1) Although the statistical technique used for automated correlation analysis (Figure 3) is robust, the voxel-wise procedure shown in Figure 3 is not very rigorous. This is not a deal-breaker because the procedure is in any case only applied within regions already found positive using the Pearson+Whitney method. Nevertheless, strictly, the voxel-wise method used is really a sort of cluster-based technique, using arbitrary thresholds for voxel-wise significance, and cluster-size threshold. Similar methods are known to be dangerously prone to false positives (see Eklund PNAS 2016). More statistically rigorous voxel-wise analysis should be presented.*

The paper cited by the reviewer (Eklund et al., "Cluster failure: Why fMRI inferences for spatial extent have inflated false-positive rates" PNAS, 113:7900-5, 2016) uses a large real-world null data set to compute empirical false positive rates for three widely-used fMRI software packages. The authors conclude: "[…] all three packages have conservative voxelwise inference and invalid clusterwise inference, for both one- and two-sample t tests. Alarmingly, the parametric methods can give a very high degree of false positives (up to 70%, compared with the nominal 5%) for clusterwise inference." Unlike the software packages evaluated in Eklund et al., our voxelwise analysis method uses a non-parametric test (Mann-Whitney *U*-test) and should not be affected by the large errors that the paper is discussing. However, to more directly address the reviewer's concerns, we have now also incorporated false discovery rate (FDR) estimations into our revised manuscript. FDR estimation is a commonly used approach for determining appropriate *p*-value thresholds in multiple testing settings. To estimate our FDR, we utilize a null data set consisting of 16+ negative control embryos (i.e. embryos that are either genetically wild-type or heterozygous for the *fezf2* mutation) for each of the 19 *in situ* probes in our library. Following in situ staining, negative controls are randomly divided into two separate groups. Since these embryos lack the *fezf2* phenotype, any differences that are detected between the two groups are presumably false positives. We then evaluate significance thresholds (*t*) over a range of *p*-values by counting the number of experimental scores (*s_exp_*; i.e. wild-type embryos vs. *fezf2* mutants) ≥ *t* and the number of null scores (*s_null_*; i.e. negative control group 1 vs. negative control group 2) ≥ *t* and calculating the FDR (*s_null_/s_exp_*). We have performed FDR analysis for both automated correlation analysis (Figure 3) and voxelwise analysis (Figure 3 and [Supplementary-material SD3-data]). The results are presented in Figure 3—figure supplement 3 and Figure 3—figure supplement 4, respectively.

Our analysis shows that we have a very small number of false positives in our negative control data and, consequently, a very low FDR. For automated correlation analysis, our FDR at the chosen *p*-value threshold (*p* < 10^-5^) is 0.043. Additionally, since the mean effect size (ES; Cohen’s *d*) is < 1.0 in the null data set for all significance thresholds evaluated, we include an additional requirement that ES must be > 1.0 for automated correlation analysis, further reducing the FDR from 0.043 to 0.020. For voxelwise analysis, our FDR at the chosen *p*-value threshold (*p* <0.5x10^-3^) is 0.004. We have substantially revised the "Automated deep-phenotyping of entire brains in zebrafish mutants" portion of the Results section to reflect these changes and to more clearly explain how *p*-value and Cohen's *d* are combined:

"We used false discovery rate (FDR) estimation, a technique for assessing false positives when conducting multiple comparisons, to set an appropriate *p*-value threshold for automated phenotyping (Noble, 2009). […] Therefore, we included an effect size requirement (*d* > 1.0) for Automated Correlation Analysis, further reducing the FDR at our selected *p*-value threshold from 0.043 to 0.020 (Figure 3—figure supplement 3)." These criteria (*p* < 10^-5^, *d* > 1.0) are used to create an Automated Correlation Analysis table depicting changes in all probes across all brain regions at 2 and 3 dpf (Figure 3, left)."

Lastly, we have removed the cluster size threshold entirely for voxelwise analysis in Figure 3 and [Supplementary-material SD3-data]. This change is motivated in part by papers such as Eklund et al., which report that cluster-based *p*-values tend to be more sensitive to statistical assumptions. Eliminating the cluster size threshold may in fact make voxelwise analysis slightly easier to visualize in our images and, in any event, has no impact on our overall analysis or conclusions.

*2) In several steps of the analysis, a large number of statistical tests are required to compare voxel data for numerous genes and brain regions. What was the basis for setting p value thresholds for so many comparisons?*

As noted in our response to the previous comment, we have now included false discovery rate (FDR) estimations for both automated correlation analysis (Figure 3 and Figure 3—figure supplement 3) and voxelwise analysis (Figure 3, Figure 3—figure supplement 4, and [Supplementary-material SD3-data]). These FDR estimations provide one basis for determining *p*-value thresholds; for both types of analysis, we have chosen a threshold that results in a FDR < 0.05. Additionally, in the case of voxelwise analysis, the main reason for generating color coded maximum intensity projections is to give readers a clear visual overview of where each gene expression pattern is increased or decreased in the *fezf2* mutant. For this purpose, it is desirable to have a FDR that is much lower than 0.05 in order to reduce background noise from false positive voxels. This can be seen in the negative control images in Figure 3—figure supplement 4, where a p-value of 0.005 gives a FDR of 0.046 but still results in maximum intensity projections with visible false positives. We have therefore used the maximum intensity projections from negative control embryos to assist us in choosing the most appropriate *p*-value threshold for voxelwise analysis (*p* <0.5x10^-3^; FDR=0.004).

*3) Please provide more detail in support of the statement "The ability to rapidly image and align thousands of WISH-stained embryos in 3D […]". Although timings for some individual steps appears in the Methods, it would be useful to summarise in the results text the rate at which lines can be processed, imaged and registered.*

We have added experimentally recorded times for all sample processing steps to the "High-throughput optical projection tomography platform" portion of the Methods section:

"Average handling time per embryo is 55.0 ± 2.9 seconds when loading and unloading is done manually (embryo loading: 19.4 ± 2.1s; image acquisition and data transfer: 18.0 ± 0.7s; embryo unloading: 17.6 ± 3.0)."

We have also clarified that alignment and reconstruction steps can be run offline (e.g. overnight, weekends, etc.) without user intervention:

"The alignment and reconstruction takes ~11 minutes per embryo to run on a 2.6 GHz hexa-core processor standard workstation and the reconstructed image has a size of 512x512x512 voxels. Both alignment and reconstruction are fully automated and can therefore be run offline without any user intervention."

*4) The authors state that the registration algorithms used in other studies (e.g. CMTK, used in Randlett, 2016) are unsuitable for their WISH data. It would be informative if they could briefly comment on why this is the case.*

Both ViBE-Z (Ronneberger et al., Nat Methods, 2012, 9:735-42) and Randlett et al. (Nat Methods, 2015, 12:1039-46) employ fluorescent confocal microscopy and use one fluorescent channel to image a background marker that clearly delineates internal morphology and a second fluorescent channel to image the expression pattern(s) of interest. In the case of ViBE-Z, the background marker is a nuclear stain, while in Randlett et al. it is anti-tERK. Both of these background stains provide sufficient internal details to allow for direct elastic registration of experimental samples to a common reference sample using the "background stain" channel. This transformation is then applied to the stain of interest in the second channel. In contrast, finding a background reference stain that highlights sufficient internal morphology to enable elastic registration, while simultaneously allowing the stain of interest to be imaged accurately and at high resolution using optical projection tomography on non-fluorescent chromogenic labels, is far more challenging. While there are various double-staining chromogenic WISH protocols that might in theory allow for a similar approach (for example: NBT/BCIP produces a dark blue stain and INT/BCIP produces a magenta stain), there are numerous practical challenges. Unlike fluorescent labels, which can be easily separated using appropriate excitation and emission filters, perfectly separating chromogenic stains of different colors into discrete channels based the transmission of visible light through a sample is much more difficult. Failure to achieve adequate separation of the two stains means that the specific gene expression pattern of interest will be partially contaminated/obscured by the background reference stain (and vice versa). Additionally, double staining chromogenic WISH protocols tend to be less robust, more time consuming, and more expensive than standard single color NBT/BCIP staining. For these reasons, existing registration algorithms and workflows are no suitable for analyzing chromogenic WISH via OPT (or at a minimum, doing so would require substantial modification and optimization of both the underlying algorithms and the WISH protocol). Finally, it should be noted that the high temperatures and stringent wash conditions characteristic of WISH protocols result in considerable change to the starting size and shape of fixed samples. These changes are relatively uniform from embryo-to-embryo, so expression patterns can still be easily aligned and compared between in situstained embryos, but aligning in situ stained embryos to an anatomical atlas or common reference frame developed from non-WISH stained embryos is more challenging and requires much greater deformation. Both ViBE-Z and Randlett et al. utilize common reference frames that are based on whole mount immunohistochemistry rather than *in situ* hybridization, introducing yet another incompatibility with chromogenic WISH stains. We have slightly modified the "registration and alignment of 3D gene expression datasets" portion of the Results section in an attempt to draw attention to these issues:

"Automated algorithms for global registration of 3D fluorescent images from zebrafish have been described (Randlett et al., 2015, Ronneberger et al., 2012), however these are unsuitable for tomographic reconstructions from chromogenic WISH as they require imaging two stains in separate fluorescence channels: 1) a detailed marker of internal morphology, which is used to stain all fish and guides deformable registration to the common reference frame and 2) the specific expression pattern of interest. […] The PRF (blue channel) is registered to the URF (green channel) using the SYTOX stain and all other embryos stained with the same probe are registered to the PRF using both the SYTOX stain and the 3D gene expression pattern (green channel)."

In addition, we have added the following details to the "in situpattern alignment" portion of the Methods section:

"When viewed in bright field (blue or green channels), SYTOX Green labels all tissues with a more or less uniform faint red counterstain. […] To overcome these challenges and align WISH-stained embryos as closely as possible to one another and to the common reference frame, we begin by a selecting a probe-specific reference for each probe in the library."

*Please make sure to submit the code and brain atlases so that they are available for download by the readers.*

We have made our source code available online at GitHub and added a "Source code" section to Methods with links:

"Core source code for image reconstruction and registration (https://github.com/aallalou/OPT-InSitu-Toolbox), along with brain atlases (Source code 1) and sample data files (Source code 2), are available online. […] Other required toolboxes that need to be downloaded and installed are the ASTRA Tomography Toolbox (van Aarle et al., 2015, Palenstijn et al., 2011) and DIPimage (Luengo Hendriks et al.)."

*Also, please consider whether you would like your article to appear as a 'Tools and Resources' article or a regular one.*

We would like our article to appear as a regular one due to our novel findings in addition to the newly introduced technology.

*Suggested revisions:*

*The authors should offer more detailed rationale for several steps in the registration procedure; additional information in the manuscript will help inform future work by other groups. Two specific questions:*

*1) For each probe, one embryo (the PRF) is selected via strongest correlation to the probe-specific iterative shape average. This embryo is then registered to the URF using a rigid+affine registration (no elastic step) for the sytox channel, then other embryos registered to the PRF using deformable registration using the green channel. But with this procedure, any inaccuracy in the PRF alignment will be transmitted to all other embryos for that probe. Because initial PRF alignment is rigid+affine and not elastic, accuracy will be limited. So why is elastic registration not used for the PRF? The final registration step (subsection “Image registration”, second paragraph) is not well explained – is this rigid+affine only? How is the average wildtype created (ISA?)*

The reviewer is correct that any minor inaccuracies in alignment of the PRF to the URF will be transmitted to all other fish stained with that in situ probe. However, this will only affect the alignment to the URF and the brain atlas. The alignment accuracy for embryos stained with the same probe (which serves as the basis for detecting alterations in gene expression) will not be affected, as they are all registered to the PRF using rigid, affine, and deformable steps. The reason for not including a deformable step when initially registering the PRF to the URF is similar to considerations we have discussed previously under “Essential revisions.” Unlike the PRF, URF does not possess any prominent internal features that can be used to guide deformable image registration; therefore, using a deformable step will not substantially improve registration accuracy. Indeed, based on our experience, using a deformable step when aligning the PRF and URF risks moving internal structures in the PRF more than desired since there are few internal features in the URF. To avoid these issues, we instead use the selection strategy described in the manuscript to identify a PRF for each in situ probe that most closely matches the "average" wild-type for that probe, under the assumption that this will give the best match to the URF (which is an average of multiple wild-type fish).

The final registration step is done using an affine transform and ensures that all embryos are aligned as correctly as possible to the common reference frame. The average wild-type used in this final step is created by calculating the average intensity of each aligned voxel in the wild-type embryos. We have clarified both of these points in the manuscript as follows:

"Once all embryos have been registered with each other, a new average wild-type is created by calculating the average intensity of each aligned voxel in the wild-type group. This average is aligned to the URF in the blue channel with an affine transform. The transformation from this final step is then applied to all wild-type and mutant embryos for that probe, ensuring they are aligned as closely as possible to the common reference frame."

*2) Using the WISH pattern to drive registration to a selected wildtype embryo means all the resulting images will have low variability, but actually mask real biological variability in individual WISH patterns. The authors acknowledge this concern at the end of the manuscript – because mutant patterns are elastically aligned to a wildtype WISH pattern differences in the mutant are minimized. My understanding is that the alternate procedure in Figure 1—figure supplement 4, that avoids this problem, is only for the 2D visualization overlays in Figure 4–Figure 5, because it is obviously not suitable for statistical comparison. The more obvious procedure would be to independently align each wildtype and mutant embryo to the URF using its sytox channel, apply the transformation to the WISH channel and then perform statistical comparisons. Why was this method not used?*

This relates to previous answers regarding the SYTOX counterstain (blue or green channels). When viewed in bright field, SYTOX labels all tissues with a more or less uniform faint red counterstain. This has the effect of highlighting surface features and a few prominent morphological landmarks (such as brain ventricles), but does not show internal brain structures in sufficient detail to allow for precise deformable registration of internal points. In contrast, the gene expression patterns in in situ stained embryos (red or green channels) provide prominent internal reference points to guide deformable registration. Therefore, independently aligning all wild-type and mutant embryos to the URF using the SYTOX (blue) channel would result in an overall decrease in alignment accuracy between embryos stained with the same probe. For the initial statistical analysis of the mutant phenotype (Figure 3), our goal was to maximize alignment precision and therefore minimize the possibility of detecting alterations in gene expression where none actually exist (false positives), even at the risk of overlooking some real biological variability (false negatives). By using the alternative alignment procedure outlined in Figure 1—figure supplement 4 – which avoids elastically aligning mutants to wild-types but is unsuitable for statistical comparisons – for subsequent 2D visualization steps, we attempt to ensure that follow-up examination of regions/probes identified by Automated Correlation Analysis will not overlook any real alterations. As noted previously, we have discussed the issues related to deformable registration of internal points based on SYTOX in greater detail in the "in situpattern alignment" portion of the Methods section:

"When viewed in bright field (blue or green channels), SYTOX Green labels all tissues with a more or less uniform faint red counterstain. […] To overcome these challenges and align WISH-stained embryos as closely as possible to one another and to the common reference frame, we begin by a selecting a probe-specific reference for each probe in the library."